# FoSR: First-order Spectral Rewiring for addressing Oversquashing in GNNs

**Kedar Karhadkar**
UCLA
kedar@math.ucla.edu

**Pradeep Kr. Banerjee**
MPI MiS
pradeep@mis.mpg.de

**Guido Montúfar**
UCLA & MPI MiS
montufar@math.ucla.edu

## Abstract

Graph neural networks (GNNs) are able to leverage the structure of graph data by passing messages along the edges of the graph. While this allows GNNs to learn features depending on the graph structure, for certain graph topologies it leads to inefficient information propagation and a problem known as oversquashing. This has recently been linked with the curvature and spectral gap of the graph. On the other hand, adding edges to the message-passing graph can lead to increasingly similar node representations and a problem known as oversmoothing. We propose a computationally efficient algorithm that prevents oversquashing by systematically adding edges to the graph based on spectral expansion. We combine this with a relational architecture, which lets the GNN preserve the original graph structure and provably prevents oversmoothing. We find experimentally that our algorithm outperforms existing graph rewiring methods in several graph classification tasks.

## 1 Introduction

Graph neural networks (GNNs) (Gori et al., 2005; Scarselli et al., 2008) are a broad class of models which process graph-structured data by passing messages between nodes of the graph. Due to the versatility of graphs, GNNs have been applied to a variety of domains, such as chemistry, social networks, knowledge graphs, and recommendation systems (Zhou et al., 2020; Wu et al., 2020). GNNs broadly follow a message-passing framework, meaning that each layer of the GNN aggregates the representations of a node and its neighbors, and transforms these features into a new representation for that node. The aggregation function used by the GNN layer is taken to be locally permutation-invariant, since the ordering of the neighbors of a node is arbitrary, and its specific form is a key component of the GNN architecture; varying it gives rise to several common GNN variants (Kipf and Welling, 2017; Veličković et al., 2018; Li et al., 2015; Hamilton et al., 2017; Xu et al., 2019). The output of a GNN can be used for tasks such as graph classification or node classification.

Although GNNs are successful in computing dependencies between nodes of a graph, they have been found to suffer from a limited capacity to capture long-range interactions. For a fixed graph, this is caused by a variety of problems depending on the number of layers in the GNN. Since graph convolutions are local operations, a GNN with a small number of layers can only provide a node with information from nodes close to itself. For a GNN with $l$ layers, the receptive field of a node (the set of nodes it receives messages from) is exactly the ball of radius $l$ about the node. For small values of $l$, this results in "underreaching", and directly limits which functions the GNN can represent. On a related note, the functions representable by GNNs with $l$ layers are limited to those computable by $l$ steps of the Weisfeiler-Lehman (WL) graph isomorphism test (Morris et al., 2019; Xu et al., 2019; Barceló et al., 2020). On the other hand, increasing the number of layers leads to its own set of problems. In contrast to other architectures that benefit from the expressivity of deeper networks, GNNs experience a decrease in accuracy as the number of layers increases (Li et al., 2018; Chen et al., 2020). This phenomenon has partly been attributed to "oversmoothing", where repeated graph convolutions eventually render node features indistinguishable (Li et al., 2018; Oono and Suzuki, 2020; Cai and Wang, 2020; Zhao and Akoglu, 2020; Rong et al., 2020; Di Giovanni et al., 2022).

Separate from oversmoothing is the problem of "oversquashing" first pointed out by Alon and Yahav (2021). As the number of layers of a GNN increases, information from (potentially) exponentially-growing receptive fields need to be concurrently propagated at each message-passing step. This leads

to a bottleneck that causes oversquashing, when an exponential amount of information is squashed into fixed-size node vectors (Alon and Yahav, 2021). Consequently, for prediction tasks relying on long-range interactions, the GNN can fail. Oversquashing usually occurs when there are enough layers in the GNN to reach any node (the receptive fields are large enough), but few enough that the GNN cannot process all of the necessary relations between nodes. Hence, for a fixed graph, the problems of underreaching, oversquashing, and oversmoothing occur in three different regimes, depending on the number of layers of the GNN.

A common approach to addressing oversquashing is to *rewire* the input graph, making changes to its edges so that it has fewer structural bottlenecks. A simple approach to rewiring is to make the last layer of the GNN fully adjacent, allowing all nodes to interact with one another (Alon and Yahav, 2021). Alternatively, one can make changes to edges of the input graph, feeding the modified graph into all layers of the GNN (Topping et al., 2022; Banerjee et al., 2022). The latter approaches can be viewed as optimizing the spectral gap of the input graph for alleviating structural bottlenecks and improving the overall quality of signal propagation across nodes (see Figure 1).

While these rewiring methods improve the connectivity of the graph, there are drawbacks to making too many modifications to the input. The most obvious problem is that we are losing out on topological information about the original graph. If the structure of the original graph is indeed relevant, adding and removing edges diminishes that benefit to the task. Another issue arises from the smoothing effects of adding edges: If we add too many edges to the input graph, an ordinary GCN will suffer from oversmoothing (Li et al., 2018). In other words, if we use this natural approach to rewiring, we experience a *trade-off between oversquashing and oversmoothing*. This observation, which does not seem to have been pointed out in earlier works, is the main motivation for the approach that we develop in this work.

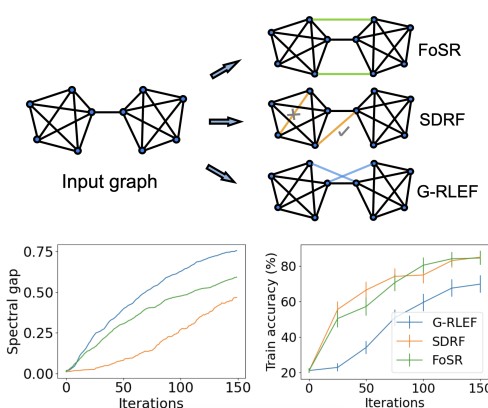

Figure 1: Top: Schematic showing different rewiring methods, FoSR (ours), SDRF (Topping et al., 2022), and G-RLEF (Banerjee et al., 2022) for alleviating structural bottlenecks in the input graph. Our method adds new edges that are labeled differently from the existing ones so that the GNN can distinguish them in training. Bottom: Normalized spectral gap and training accuracy as functions of the number of rewiring iterations for a learning task modeled on the NEIGHBORSMATCH problem for a path-of-cliques input (for details, see Appendix B.1.1).

## 1.1 MAIN CONTRIBUTIONS

This paper presents a new framework for rewiring a graph to reduce oversquashing in GNNs while preventing oversmoothing. Here are our main contributions:

- We introduce a framework for graph rewiring which can be used with any rewiring method that sequentially adds edges. In contrast to previous approaches that only modify the input graph (e.g., Topping et al., 2022; Banerjee et al., 2022; Bober et al., 2022), our solution gives special labels to the added edges. We then use a relational GNN on this new graph, with the relations corresponding to whether the edge was originally in the input graph or added during the rewiring. This allows us to preserve the input graph topology while using the new edges to improve its connectivity. In Theorem 3 we show that this approach also prevents oversmoothing.

- We introduce a new rewiring method, FoSR (First-order Spectral Rewiring) aimed at optimizing the spectral gap of the graph input to the GNN (Algorithm 1). This algorithm computes the first-order change in the spectral gap from adding each edge, and then adds the edge which maximizes this (Theorem 4 and Proposition 5).

- We empirically demonstrate that the proposed method results in faster spectral expansion (a marker of reduced oversquashing) and improved test accuracy against several baselines on several graph

classification tasks (see Table 1). Experiments demonstrate that the relational structure preserving the original input graph significantly boosts test accuracy.

## 1.2 RELATED WORKS

Past approaches to reducing oversquashing have hinged upon choosing a measure of oversquashing, and modifying the edges of the graph to minimize it. Topping et al. (2022) argue that negatively curved edges are responsible for oversquashing drawing on curvature notions from Forman (2003) and Ollivier (2009). They introduce a rewiring method known as *stochastic discrete Ricci Flow* (SDRF), which aims to increase the balanced Forman curvature of negatively curved edges by adding new edges. Bober et al. (2022) extend this line of investigation by considering the same type of rewiring but using different notions of discrete curvature. Banerjee et al. (2022) approach oversquashing from an information-theoeretic viewpoint, measuring it in terms of the spectral gap of the graph and demonstrate empirically that this can increase accuracy for certain graph classification tasks. They propose a rewiring algorithm *greedy random local edge flip* (G-RLEF) motivated by an expander graph construction employing an effective resistance (Lyons and Peres, 2017) based edge sampling strategy. The work of Alon and Yahav (2021) first pointing at oversquashing also introduced an approach to rewiring, where they made the last GNN layer an expander – the complete graph that allows every pair of nodes to connect to each other. They also experimented with making the last layer partially adjacent (randomly including any potential edge). This can be thought of as a form of spectral expansion in the final layer since random graphs have high spectral gap (Friedman, 1991). In contrast to these works, our method gives a practical way of achieving the largest possible increase in the spectral graph with the smallest possible modification of the input graph and in fact preserving the input graph topology via a relational structure.

Although not as closely related, we find it worthwhile also pointing at following works in this general context. Prior to the diagnosis of the oversquashing problem, Klicpera et al. (2019) used graph diffusion to rewire the input graph, improving long-range connectivity for the GNN. Rewiring can also be performed while training a GNN. Arnaiz-Rodríguez et al. (2022) use first-order spectral methods to define a loss function depending on the adjacency matrix, allowing a GNN to learn a rewiring that alleviates oversquashing. We should mention that aside from rewiring the input graph, some works pursue different approaches to solve oversquashing, such as creating positional embeddings for the nodes or edges inspired by the transformer architecture (Vaswani et al., 2017). The most direct generalization of this approach to graphs is using Laplacian embeddings (Kreuzer et al., 2021; Dwivedi and Bresson, 2020). Brüel-Gabrielsson et al. (2022) combine this with adding neighbors to encode the edges which are the result of multiple hops.

## 2 PRELIMINARIES

### 2.1 BACKGROUND ON SPECTRAL GRAPH THEORY

Let $\mathcal{G} = (\mathcal{V}, \mathcal{E}, \mathcal{R})$ be an undirected graph with node set $\mathcal{V}$, $|\mathcal{V}| = n$, edge set $\mathcal{E}$, $|\mathcal{E}| = m$, and relation set $\mathcal{R}$. The set $\mathcal{R}$ is a finite set of relation types, and elements $(u, v, r) \in \mathcal{E}$ consist of a pair of nodes $u, v \in \mathcal{V}$ together with an associated relation type $r \in \mathcal{R}$. When the relation type of an edge is not relevant, we will simply write $(u, v)$ for an edge. For each $v \in \mathcal{V}$ we define $\mathcal{N}(v)$ to consist of all neighbors of $v$, that is all $u \in \mathcal{V}$ such that there exists an edge $(u, v) \in \mathcal{E}$. For each $r \in \mathcal{R}$ and $v \in \mathcal{V}$, we define $\mathcal{N}_r(v)$ to consist of all neighbors of $v$ of relation type $r$. The degree $d_v$ of a node $v \in \mathcal{V}$ is the number of neighbors of $v$. We define the adjacency matrix $A = A(\mathcal{G})$ by $A_{ij} = 1$ if $(i, j) \in \mathcal{E}$, and $A_{ij} = 0$ otherwise. Let $D = D(\mathcal{G})$ denote the diagonal matrix of degrees given by $D_{ii} = d_i$. The normalized Laplacian $L = L(\mathcal{G})$ is defined as $L = I - D^{-1/2}AD^{-1/2}$. We will often add self-loops (edges $(i, i)$ for $i \in \mathcal{V}$) to the graphs we consider, so we define augmented versions of the above matrices corresponding to graphs with self-loops added. If $\mathcal{G}$ is a graph without self-loops, we define its augmented adjacency matrix $\tilde{A} := I + A$, its augmented degree matrix $\tilde{D} = I + D$, and its augmented Laplacian $\tilde{L} = I - \tilde{D}^{-1/2}\tilde{A}\tilde{D}^{-1/2}$.

We denote the eigenvalues of the normalized Laplacian $L$ by $0 = \lambda_1 \leq \lambda_2 \leq \cdots \leq \lambda_n \leq 2$. Let $\mathbf{1}$ denote the constant function which assumes the value 1 on each node. Then $D^{1/2}\mathbf{1}$ is an eigenfunction of $L$ with eigenvalue 0. The *spectral gap* of $\mathcal{G}$ is $\lambda_2 - \lambda_1 = \lambda_2$. We say that $\mathcal{G}$ has

good spectral expansion if it has a large spectral gap. In Appendix A, we review the relation between the spectral gap and a related measure of graph expansion, the *Cheeger constant*.

## 2.2 BACKGROUND ON RELATIONAL GNNS

The framework we propose fundamentally relies on relational GNNs (R-GNNs) (Battaglia et al., 2018), so we review their formulation here. We define a general R-GNN layer by

$$h_v^{(k+1)} = \phi_k\Big(h_v^{(k)}, \sum_{r \in \mathcal{R}} \sum_{u \in \mathcal{N}_r(v)} \psi_{k,r}(h_u^{(k)}, h_v^{(k)})\Big),$$

where the $\psi_{k,r} : \mathbb{R}^{d_k} \times \mathbb{R}^{d_k} \to \mathbb{R}^{d_{k+1}}$ are *message-passing functions*, and the $\phi_k : \mathbb{R}^{d_{k+1}} \to \mathbb{R}^{d_{k+1}}$ are *update functions*. The $\phi_k$ and $\psi_{k,r}$ can either be fixed mappings or learned during training. All GNN layer types we consider will be special cases of this general R-GNN layer. We recover a classical (non-relational) GNN when there is only one relation, $|\mathcal{R}| = 1$.

As a special case of R-GNNs, R-GCN layers are defined by the update

$$h_v^{(k+1)} = \sigma\Big(W^{(k)} h_v^{(k)} + \sum_{r \in \mathcal{R}} \sum_{u \in \mathcal{N}_r(v)} \frac{1}{c_{u,v}} W_r^{(k)} h_u^{(k)}\Big),$$

where $\sigma$ is a nonlinear activation, $c_{u,v}$ is a normalization factor, and $W^{(k)}$, $W_r^{(k)}$ are relation-specific learned linear maps. One can interpret the $W h_v^{(k)}$ term as adding self-loops to the original graph, and encoding these self-loops as their own relation. We often take $\sigma = \text{ReLU}$, and $c_{u,v} = \sqrt{(1 + d_u)(1 + d_v)}$.

Other specializations of R-GNNs include graph convolutional networks (GCNs) (Kipf and Welling, 2017), defined by

$$h_v^{(k+1)} = \sigma\Big(\sum_{u \in \mathcal{N}(v) \cup \{v\}} \frac{1}{c_{u,v}} W^{(k)} h_u^{(k)}\Big),$$

and graph isomorphism networks (GINs) (Xu et al., 2019), defined by

$$h_v^{(k+1)} = \text{MLP}^{(k)}\Big(\sum_{u \in \mathcal{N}(v) \cup \{v\}} h_u^{(k)}\Big).$$

## 3 RELATIONAL REWIRING OF GNNS

We introduce a graph rewiring framework to improve connectivity while retaining the original graph via a relational structure, which demonstrably allows us to also control the *rate of smoothing*. The rate of smoothing measures the similarity of neighboring node features in the GNN output graph. Adding separate weights for the rewired edges allows the network more flexibility in learning an appropriate rate of smoothing. Our main result in this section, Theorem 3 makes this precise.

## 3.1 RELATIONAL REWIRING

We incorporate relations into our architecture in the following way. Suppose that we rewire $\mathcal{G} = (\mathcal{V}, \mathcal{E}_1)$ by adding edges, yielding a rewired graph $\mathcal{G}' = (\mathcal{V}, \mathcal{E}_1 \cup \mathcal{E}_2)$. We equip $\mathcal{G}'$ with a relational structure by assigning each edge in $\mathcal{E}_1$ the edge type 1, and each edge in $\mathcal{E}_2$ the edge type 2. For example, in an R-GCN, this relational structure would result in the following layer form:

$$h_v^{(k+1)} = \sigma\Big(W^{(k)} h_v^{(k)} + \sum_{(u,v) \in \mathcal{E}_1} \frac{1}{c_{u,v}} W_1^{(k)} h_u^{(k)} + \sum_{(u,v) \in \mathcal{E}_2} \frac{1}{c_{u,v}} W_2^{(k)} h_u^{(k)}\Big).$$

The original layer (before relational rewiring) would include only the first two terms. A rewired graph with no relational structure would add the third term but use the same weights $W_2^{(k)} = W_1^{(k)}$.

A relational structure serves to provide the GNN with more flexibility, since it remembers both the original structure of the graph and the rewired structure. In the worst case scenario, if we add too many edges, the GNN may counterbalance this by setting most of the weights $W_2^{(k)}$ to be 0, focusing its attention on the original graph. In a more realistic scenario, the GNN can use the original edge weights $W_1^{(k)}$ to compute important structural features of the graph, and the weights $W_2^{(k)}$ simply to transmit information along the graph. Since the weights assigned to the original and rewired edges are separate, their purposes can be served in parallel without sacrificing the topology of the graph. Finally, this R-GCN model allows us to better regulate the rate of smoothing as we demonstrate next.

## 3.2 Rate of smoothing for R-GCN with rewiring

In this section, we analyze the smoothing effects of repeatedly applying R-GCN layers with rewiring. Here, we consider R-GCN layers $\varphi : \mathbb{R}^{n \times p} \to \mathbb{R}^{n \times p}$ of the form

$$\varphi(X) = X\Theta + \sum_{r \in \mathcal{R}} D^{-1/2} A_r D^{-1/2} X \Theta_r, \tag{1}$$

where $\Theta, \Theta_r \in \mathbb{R}^{p \times p}$ are weight matrices, $A_r$ is the adjacency matrix of the $r$-th relation, and $D$ is the matrix of degrees of $\mathcal{G}$. For a vanilla GCN without relations or residual connections, one problem with rewiring the graph is that adding too many edges will result in oversmoothing (Di Giovanni et al., 2022). Indeed, adding edges to a graph will by definition increase its Cheeger constant. For input graphs with a high spectral gap (and hence high Cheeger constant), a GCN will produce similar representations for each node (Xu et al., 2018; Oono and Suzuki, 2020). As an extreme case, if we replace the graph with a complete graph, the GCN layer will assign each node the same representation. We will show that R-GCNs are robust to this effect, allowing us to add edges without the adverse side effect of oversmoothing.

**Definition 1.** Let $\mathcal{G}$ be a connected graph with adjacency matrix $A$ and normalized Laplacian $L$. For $i \in \mathcal{V}$, let $d_i$ denote the degree of node $i$. Given a scalar field $f \in \mathbb{R}^n$, its *Dirichlet energy* with respect to $\mathcal{G}$ is defined as

$$\mathscr{E}(f) := \tfrac{1}{2} \sum_{i,j} A_{i,j} \left( \frac{f_i}{\sqrt{d_i}} - \frac{f_j}{\sqrt{d_j}} \right)^2 = f^T L f.$$

For a vector field $X \in \mathbb{R}^{n \times p}$, we define

$$\mathscr{E}(X) := \tfrac{1}{2} \sum_{i,j,k} A_{i,j} \left( \frac{X_{i,k}}{\sqrt{d_i}} - \frac{X_{j,k}}{\sqrt{d_j}} \right)^2 = \mathrm{Tr}(X^T L X).$$

The Dirichlet energy of a unit-norm function on the nodes of a graph is a measure of how "non-smooth" it is (Chung and Graham, 1997). The Dirichlet energy of a function $f$ is small when $f_i/\sqrt{d_i}$ is close in value to $f_j/\sqrt{d_j}$ for all edges $(i, j)$. In the case where $\mathcal{G}$ is $d$-regular (all nodes have degree $d$), this reduces to the familiar notion of adjacent nodes having representations close in value. Hence, we make the following definition:

**Definition 2.** Let $\mathcal{G}$ be a graph and $\varphi : \mathbb{R}^{n \times p} \to \mathbb{R}^{n \times p}$ be a mapping. We define the *rate of smoothing* of $\varphi$ with respect to $\mathcal{G}$ as

$$RS_{\mathcal{G}}(\varphi) := 1 - \left( \frac{\sup_{X : \mathscr{E}(X) \neq 0} \mathscr{E}(\varphi(X))/\mathscr{E}(X)}{\sup_{X : X \neq 0} \|\varphi(X)\|_F^2 / \|X\|_F^2} \right)^{1/2}.$$

The numerator of the fraction above indicates the rate of decay (or expansion) of the Dirichlet energy upon applying $\varphi$. We take a supremum over $X$ to find the largest possible relative change in energy upon applying $\varphi$. We would also like our notion of the rate of smoothing to be scale-invariant; multiplying $\varphi$ by a scalar should not change its rate of smoothing. To impose scale-invariance, we divide by a factor in the denominator which captures how much $\varphi$ scales up the entries of $X$. By defining the the rate of smoothing as a ratio of two norms, we can estimate them separately which gives us a good theoretical handle of smoothing. Note that if $\varphi$ is linear,

$$RS_{\mathcal{G}}(\varphi) := 1 - \left( \frac{\sup_{\mathscr{E}(X)=1} \mathscr{E}(\varphi(X))}{\sup_{\|X\|_F=1} \|\varphi(X)\|_F^2} \right)^{1/2},$$

since $\| \cdot \|_F^2$ and $\mathscr{E}$ are quadratic forms. The following theorem shows that R-GCN layers are flexible in their ability to choose an appropriate rate of smoothing for a graph:

**Theorem 3.** *Let $\mathcal{G}_1 = (\mathcal{V}, \mathcal{E}_1)$ be a graph and $\mathcal{G}_2 = (\mathcal{V}, \mathcal{E}_1 \cup \mathcal{E}_2)$ be a rewiring of $\mathcal{G}_1$. Consider an R-GCN layer $\varphi$ defined as in (1), with relations $r_1 = \mathcal{E}_1$, $r_2 = \mathcal{E}_2$. Then for any $\lambda \in [0, \lambda_2(L(\mathcal{G}_2))]$, there exist values of $\Theta, \Theta_1, \Theta_2$ for which $\varphi$ smooths with rate $RS_{\mathcal{G}_2}(\varphi) = \lambda$ with respect to $\mathcal{G}_2$.*

*Proof.* The map $\varphi$ is given by

$$\varphi(X; \Theta, \Theta_1, \Theta_2) = X\Theta + D^{-1/2} A_1 D^{-1/2} X \Theta_1 + D^{-1/2} A_2 D^{-1/2} X \Theta_2,$$

where $A = A_1 + A_2$ is the adjacency matrix of $\mathcal{G}_2$. Here $D$ is the degree matrix of $\mathcal{G}_2$. Fix $\alpha \in [0, 1]$, and take $\Theta_1 = \Theta_2 = \alpha I$ and $\Theta = (1 - \alpha)I$. Then

$$\varphi(X) = (1 - \alpha)X + \alpha D^{-1/2}A_1D^{-1/2}X + \alpha D^{-1/2}A_2D^{-1/2}X$$
$$= (1 - \alpha)X + \alpha D^{-1/2}AD^{-1/2}X = (I - \alpha L)X,$$

where $L$ is the normalized Laplacian of $\mathcal{G}_2$. We will show that $\varphi$ smooths with rate $\alpha\lambda_2(L)$. Let $L = U\Sigma U^{-1}$ be an orthogonal diagonalization of $L$. Let $\lambda_1, \dots, \lambda_n$ be the eigenvalues of $L$ in ascending order and recall that $0 \le \lambda_i \le 2$ for all $i$ with $\lambda_1 = 0$. We have

$$\mathscr{E}(\varphi(X)) = \text{Tr}(X^T L(I - \alpha L)^2 X) = \text{Tr}(X^T U\Sigma(I - \alpha\Sigma)^2 U^T X) = \sum_{i,j} \lambda_i(1 - \alpha\lambda_i)^2((U^T X)_{i,j})^2$$

$$\overset{(a)}{\le} \sum_{i,j} \lambda_i(1 - \alpha\lambda_2)^2((U^T X)_{i,j})^2 = (1 - \alpha\lambda_2)^2 \text{Tr}(X^T U\Sigma U^T X) = (1 - \alpha\lambda_2)^2 \mathscr{E}(X),$$

with equality in (a) when $(U^T X)_{i,j} = 0$ for $i \ne 2$. Hence,

$$\sup_{\mathscr{E}(X) \ne 0} \frac{\mathscr{E}(\varphi(X))}{\mathscr{E}(X)} = (1 - \alpha\lambda_2)^2. \tag{2}$$

Next, we compute

$$\|\varphi(X)\|_F^2 = \text{Tr}(X^T(I - \alpha L)^2 X) = \text{Tr}(X^T U(I - \alpha\Sigma)^2 UX)$$

$$= \sum_{i,j}(1 - \alpha\lambda_i)^2((UX)_{i,j})^2 \overset{(b)}{\le} \sum_{i,j}((UX)_{i,j})^2 = \|UX\|_F^2 = \|X\|_F^2,$$

with equality in (b) when $(UX)_{i,j} = 0$ for $i \ne 1$. Hence

$$\sup_{X \ne 0} \frac{\|\varphi(X)\|_F^2}{\|X\|_F^2} = 1. \tag{3}$$

Combining (2) and (3) yields $RS(\varphi) = \alpha\lambda_2 = \alpha\lambda_2(L)$. Since this holds for any $\alpha \in [0, 1]$, the result follows. $\square$

In Appendix B.3, we empirically demonstrate that R-GNNs achieve lower Dirichlet energy than GNNs.

## 4 FoSR: First order Spectral Rewiring

In order to use the R-GCN framework, we need to make a choice of rewiring algorithm that is capable of adding edges without removing any. We propose a first-order spectral rewiring algorithm (FoSR) with the goal of improving graph connectivity. The *rate of spectral expansion*, i.e., the rate at which the spectral gap improves as a function of the rewiring iterations is one of the key determinants of oversquashing (Banerjee et al., 2022; Deac et al., 2022). Compared to existing approaches such as SDRF (Topping et al., 2022) and G-RLEF (Banerjee et al., 2022), the proposed algorithm has a faster rate of spectral expansion. See Figure 1 for an illustration.

Our algorithm FoSR follows a similar philosophy as Chan and Akoglu (2016) optimizing the spectral gap of the input graph by sequentially adding edges such that the rate of spectral expansion is maximized. At each step, we wish to add the edge that maximizes the spectral gap of the resulting graph. However, computing the spectral gap for each edge addition is expensive as it requires us to compute eigenvalues for $O(n^2)$ different matrices. Instead of computing this quantity directly, we use a first-order approximation of the spectral gap based on matrix perturbation theory. The following theorem determines the first-order change of the eigenvalues of a perturbed symmetric matrix:

**Theorem 4.** *For symmetric matrices $M \in \mathbb{R}^{n \times n}$ with distinct eigenvalues, the $i$-th largest eigenvalue $\lambda_i$ satisfies*

$$\nabla_M \lambda_i(M) = x_i x_i^T,$$

*where $x_i$ denotes the (normalized) eigenvector for the $i$-th largest eigenvalue of $M$.*

We provide a proof in Appendix C. Stated differently, the previous theorem allows us to make the linear approximation

$$\lambda_2(M + \delta M) \approx \lambda_2(M) + \text{Tr}(\nabla \lambda_2^T (\delta M)) = \lambda_2(M) + x_2^T (\delta M) x_2. \tag{4}$$

Let us apply the approximation (4) to the spectrum of a graph $\mathcal{G}$. We wish to sequentially add an edge $(u, v)$ in a way that maximizes the spectral gap of the resulting graph. Equivalently, the second-largest eigenvalue of $D^{-1/2} A D^{-1/2}$ should be minimized, since this matrix is equal to $I - L$. Using Theorem 4 for the matrix $D^{-1/2} A D^{-1/2}$, we obtain the following result.

**Proposition 5.** *The first-order change in $\lambda_2 = \lambda_2(D^{-1/2} A D^{-1/2})$ from adding the edge $(u, v)$ is*

$$\frac{2x_u x_v}{(\sqrt{1 + d_u})(\sqrt{1 + d_v})} + 2\lambda_2 x_u^2 \left( \frac{\sqrt{d_u}}{\sqrt{1 + d_u}} - 1 \right) + 2\lambda_2 x_v^2 \left( \frac{\sqrt{d_v}}{\sqrt{1 + d_v}} - 1 \right), \tag{5}$$

*where $x$ denotes the second eigenvector of $D^{-1/2} A D^{-1/2}$, and $x_u$ denotes the $u$-th entry of $x$.*

We provide a proof in Appendix C. We choose to minimize the first term in (5)

$$\frac{2x_u x_v}{\sqrt{(1 + d_u)(1 + d_v)}}, \tag{6}$$

since this is simpler and cheaper to compute. To see why this approximation is reasonable, note that $\frac{1}{(\sqrt{1+d_u})(\sqrt{1+d_v})} \sim (d_u d_v)^{-1/2}$ as $d_u, d_v \to \infty$, while $\frac{\sqrt{d_u}}{\sqrt{1+d_u}} - 1 \sim d_u^{-2}$. So if the degrees of the nodes are sufficiently large and comparable in size, the first term in (5) dominates. Again, our decision to optimize the first term is chosen for computational reasons. In Appendix B.2, we show that FoSR has significantly smaller computation overhead compared to SDRF on real and synthetic datasets. Further, we show in Appendix B.1.2 that FoSR has a much faster rate of spectral expansion compared to SDRF. Finally in Appendix B.4, we discuss the approximation error of FoSR.

For each iteration that it runs, FoSR computes an approximation of the second eigenvector $x \in \mathbb{R}^n$ of $D^{-1/2} A D^{-1/2}$ and chooses an edge $(u, v)$ which minimizes (6). To produce an initial approximation of $x$, we use power iteration (rather than computing a full eigendecomposition). Let $d \in \mathbb{R}^n$ denote the vector of degrees of $\mathcal{G}$. Let $\sqrt{d}$ denote the entry-wise square root of $d$, i.e., with components $\sqrt{d_i}$. Recall that $\sqrt{d}$ is the first eigenvector of $D^{-1/2} A D^{-1/2}$. Since $x$ is the second eigenvector of $D^{-1/2} A D^{-1/2}$, we may approximate it by repeatedly applying $D^{-1/2} A D^{-1/2}$ and then subtracting the component in the direction of the first eigenvector. This map is given by

$$x \mapsto D^{-1/2} A D^{-1/2} x - \frac{\langle x, \sqrt{d} \rangle}{2m} \sqrt{d}.$$

After applying this map, we normalize $x$ back to a unit vector, $x \mapsto \frac{x}{\|x\|}$. FoSR operates by repeatedly alternating between adding an edge and computing a new approximation of $x$ via power iteration. Importantly, our algorithm avoids computing a full eigendecomposition of $\mathcal{G}$ by only approximating its second eigenvector. We compute increasingly accurate estimates of this eigenvector using power iteration. The steps of our method are outlined in Algorithm 1.

---

**Algorithm 1 FoSR: First-order Spectral Rewiring**

    **Input**: $\mathcal{G} = (\mathcal{V}, \mathcal{E})$, iteration count $k$, initial number of power iterations $r$
    **Output**: Rewired graph $\mathcal{G}' = (\mathcal{V}, \mathcal{E}')$
 1: Initialize $x \in \mathbb{R}^n$ arbitrarily
 2: **for** $i = 1, 2, \cdots, r$ **do**
 3:      $x \leftarrow D^{-1/2} A D^{-1/2} x - \frac{\langle x, \sqrt{d} \rangle}{2m} \sqrt{d}$      $\triangleright$ Approximate second eigenvector before rewiring
 4:      $x \leftarrow \frac{x}{\|x\|_2}$
 5: **end for**
 6: **for** $i = 1, 2, \cdots, k$ **do**
 7:      Add edge $(i, j)$ which minimizes $\frac{x_i x_j}{\sqrt{(1 + d_i)(1 + d_j)}}$
 8:      $x \leftarrow D^{-1/2} A D^{-1/2} x - \frac{\langle x, \sqrt{d} \rangle}{2m} \sqrt{d}$      $\triangleright$ Power iteration to update second eigenvector
 9:      $x \leftarrow \frac{x}{\|x\|_2}$
10: **end for**

---

The number of edge additions $k$ and the initial number of power iterations $r$ are hyperparameters. The number of power iterations $r$ only needs to be chosen large enough to produce an initial approximation of the eigenvector $x$. In practice, we found that taking $r$ between 5 and 10 is sufficient. The proper choice of iteration count $k$ is problem-specific, and can be chosen to be such that the spectral gap increases sufficiently. We tried multiple configurations of $k$ when training models.

**Computational complexity**   Taking advantage of the sparsity of $\mathcal{G}$, FoSR requires $O(m)$ floating-point operations for each power iteration and $O(n^2)$ operations for each edge added. The $O(n^2)$ operations come from searching over all node pairs $(i, j)$ for the minimal value of $y_i y_j$, where $y_i = x_i / \sqrt{1 + d_i}$. This results in a cost of $O(kn^2)$. When the graph is sparse, we can often compute the minimal value of $y_i y_j$ in a faster way. We can choose $i = \arg\min_k y_k$ and $j = \arg\max_k y_k$ if $\min_k y_k \leq 0$ and the $(i, j)$ chosen in this way is not already an edge. Indeed, the probability of $(i, j)$ not already being an edge is higher if $G$ is sparse. In this case, the edge addition only takes $O(m)$ operations. More generally, we can relax the minimization by choosing $i = \arg\min_k y_k$ and $j = \arg\max_{k \notin \mathcal{N}(i)} y_k$ when $\min_k y_k \leq 0$. We can use a similar relaxation when $\min_k y_k > 0$; take $i = \arg\min_k y_k$ and $j = \arg\min_{k \notin \mathcal{N}(i)} y_k$. If we use the above relaxations, the total cost is $O(km)$.

## 5   EXPERIMENTS

We compare our rewiring methods to existing ones to demonstrate the efficacy of relational rewiring as well as rewiring informed by spectral expansion on several graph classification tasks.

**Datasets**   We consider graph classification tasks REDDIT-BINARY, IMDB-BINARY, MUTAG, ENZYMES, PROTEINS, COLLAB from the TUDataset (Morris et al., 2020) where the topology of the graphs in relation to the task has been identified to require long-range interactions. While certain node classification tasks have also been considered in previous works, these have been found to be tractable with nearest neighbor information (Brockschmidt, 2020).

**Compared methods**   We focus on approaches that preprocess the input graph by adding edges. Diffusion Improves Graph Learning (DIGL) (Klicpera et al., 2019) is a diffusion-based rewiring scheme that computes a kernel evaluation of the adjacency matrix, followed by sparsification. SDRF (Topping et al., 2022) surgically rewires a graph by adding edges to support other edges with low curvature, which are locations of bottlenecks. For SDRF, we include results for both the original method (configured to only add edges), and our relational method (again only add edges, and include the added edges with their own relation). Fully adjacent layers (Alon and Yahav, 2021) rewire the graph by adding all possible edges, setting $\mathcal{E}_2 = (\mathcal{V} \times \mathcal{V}) \setminus \mathcal{E}_1$. We include results for rewiring only the last layer (last layer FA) and rewiring every layer (every layer FA).

**Experimental details**   We test each rewiring algorithm on an R-GCN (Schlichtkrull et al., 2018), with the relations varying depending on the rewiring method. For rewirings which do not explicitly assign relations to edges, we use two relations: one for the default edge type, and one for self-loops. In other words, we simply use the rewired graph and add residual connections to the architecture. For each task, we use the same GNN architecture for all rewiring methods to illustrate how the choice of rewiring affects test performance. That is, we hold the number of layers, hidden dimension, and dropout probability constant across all rewiring methods for a given dataset. The hyperparameter values are reported in Appendix D.1. However, for each dataset, we separately tune all hyperparamters specific to the rewiring method (such as the number of iterations for FoSR and SDRF). We train all models with an Adam optimizer with a learning rate of $10^{-3}$ and a scheduler which reduces the learning rate by a factor of 10 after 10 epochs with no improvement in validation loss. For FoSR and SDRF, we only tuned the number of rewiring iterations. For DIGL (Klicpera et al., 2019), we tuned the sparsification threshold ($\epsilon$) and teleport probability ($\alpha$).

For optimizing hyperparamters and evaluating each configuration, we first select a test set consisting of 10 percent of the graphs and a development set consisting of the other 90 percent of the graphs. We determine accuracies of each configuration using 100 random train/validation splits of the development set consisting of 80 and 10 percent of the graphs, respectively. When training the GNNs, we use a stopping patience of 100 epochs based on the validation loss. We evaluate each hyperparameter configuration using the validation accuracy, so the test set is only used once to

generate results for the best hyperparamter configurations. For the test results, we record 95 percent confidence intervals for the hyperparameters with the best validation accuracy using the 100 runs.

**Results** Table 1 shows the results of our experiments on various GNN layer types. Shown is the test accuracy. For each rewiring method, using a relational GNN type improves classification accuracy. The boost in accuracy from using an R-GNN is greatest for the rewiring methods which surgically add edges (FoSR and SDRF). Out of all of the rewiring methods, FoSR typically achieves the best classification accuracy when relational rewiring is used. When relational rewiring is not used, FoSR still outperforms DIGL and SDRF for most datasets. We note that +FA is sometimes competitive, but this is highly dependent on the GNN architecture. In particular, it does not perform well for GCNs and R-GCNs. Our results are also very competitive against results reported for the DiffWire methods by Arnaiz-Rodríguez et al. (2022, Table 1), although they use a different type of architecture that is not directly comparable to preprocessing methods like ours or SDRF which sequentially add edges.

Table 1: Results of rewiring methods for GCN and GIN comparing standard and relational. The best results in each setting are highlighted in bold font and best across settings are highlighted red.

**GCN**

| Rewiring | REDDIT-BINARY | IMDB-BINARY | MUTAG | ENZYMES | PROTEINS | COLLAB |
|---|---|---|---|---|---|---|
| None | $68.255 \pm 1.098$ | $49.770 \pm 0.817$ | $72.150 \pm 2.442$ | $27.667 \pm 1.164$ | $70.982 \pm 0.737$ | $33.784 \pm 0.488$ |
| Last layer FA | $68.485 \pm 0.945$ | $48.980 \pm 0.945$ | $70.050 \pm 2.027$ | $26.467 \pm 1.204$ | $71.018 \pm 0.963$ | $33.320 \pm 0.435$ |
| Every layer FA | $48.490 \pm 1.044$ | $48.170 \pm 0.801$ | $70.450 \pm 1.960$ | $18.333 \pm 1.038$ | $60.036 \pm 0.925$ | $\mathbf{51.798} \pm \mathbf{0.419}$ |
| DIGL | $49.980 \pm 0.680$ | $\mathbf{49.910} \pm 0.841$ | $71.350 \pm 2.391$ | $27.517 \pm 1.053$ | $70.607 \pm 0.731$ | $15.530 \pm 0.294$ |
| SDRF | $68.620 \pm 0.851$ | $49.400 \pm 0.904$ | $71.050 \pm 1.872$ | $\mathbf{28.367} \pm 1.174$ | $70.920 \pm 0.792$ | $33.448 \pm 0.472$ |
| FoSR | $\mathbf{70.330} \pm 0.727$ | $49.660 \pm 0.864$ | $\mathbf{80.000} \pm 1.574$ | $25.067 \pm 0.994$ | $\mathbf{73.420} \pm 0.811$ | $33.836 \pm 0.584$ |

**R-GCN**

| Rewiring | REDDIT-BINARY | IMDB-BINARY | MUTAG | ENZYMES | PROTEINS | COLLAB |
|---|---|---|---|---|---|---|
| None | $49.850 \pm 0.653$ | $50.012 \pm 0.917$ | $69.250 \pm 2.085$ | $28.600 \pm 1.186$ | $69.518 \pm 0.725$ | $33.602 \pm 1.047$ |
| Last layer FA | $49.800 \pm 0.626$ | $50.650 \pm 0.964$ | $70.550 \pm 1.810$ | $28.233 \pm 1.138$ | $69.527 \pm 0.815$ | $34.732 \pm 1.194$ |
| Every layer FA | $49.950 \pm 0.593$ | $50.500 \pm 0.891$ | $70.500 \pm 1.836$ | $33.400 \pm 1.142$ | $71.670 \pm 0.882$ | $33.616 \pm 0.978$ |
| DIGL | $49.995 \pm 0.619$ | $49.670 \pm 0.843$ | $73.400 \pm 2.007$ | $28.283 \pm 1.213$ | $68.232 \pm 0.851$ | $16.926 \pm 1.441$ |
| SDRF | $58.620 \pm 0.647$ | $53.640 \pm 1.043$ | $72.300 \pm 2.215$ | $33.483 \pm 1.245$ | $69.107 \pm 0.759$ | $67.990 \pm 0.386$ |
| FoSR | $\mathbf{76.590} \pm 0.531$ | $\mathbf{64.050} \pm 1.123$ | $\mathbf{84.450} \pm 1.517$ | $\mathbf{35.633} \pm 1.151$ | $\mathbf{73.795} \pm 0.692$ | $\mathbf{70.650} \pm 0.482$ |

**GIN**

| Rewiring | REDDIT-BINARY | IMDB-BINARY | MUTAG | ENZYMES | PROTEINS | COLLAB |
|---|---|---|---|---|---|---|
| None | $86.785 \pm 1.056$ | $70.180 \pm 0.992$ | $77.700 \pm 3.602$ | $33.800 \pm 1.115$ | $70.804 \pm 0.827$ | $72.992 \pm 0.384$ |
| Last layer FA | $\color{red}\mathbf{90.220} \pm 0.4750$ | $70.910 \pm 0.788$ | $\mathbf{83.450} \pm 1.742$ | $47.400 \pm 1.387$ | $72.304 \pm 0.666$ | $\mathbf{75.056} \pm \mathbf{0.406}$ |
| Every layer FA | $50.360 \pm 0.648$ | $49.160 \pm 0.870$ | $72.550 \pm 3.016$ | $28.383 \pm 1.052$ | $70.375 \pm 0.910$ | $32.894 \pm 0.390$ |
| DIGL | $76.035 \pm 0.774$ | $64.390 \pm 0.907$ | $79.700 \pm 2.150$ | $35.717 \pm 1.198$ | $70.759 \pm 0.774$ | $54.504 \pm 0.410$ |
| SDRF | $86.440 \pm 0.590$ | $69.720 \pm 1.152$ | $78.400 \pm 2.803$ | $\mathbf{35.817} \pm 1.094$ | $69.813 \pm 0.792$ | $72.958 \pm 0.419$ |
| FoSR | $87.350 \pm 0.598$ | $\mathbf{71.210} \pm 0.919$ | $78.000 \pm 2.217$ | $29.200 \pm 1.376$ | $\color{red}\mathbf{75.107} \pm 0.817$ | $73.278 \pm 0.416$ |

**R-GIN**

| Rewiring | REDDIT-BINARY | IMDB-BINARY | MUTAG | ENZYMES | PROTEINS | COLLAB |
|---|---|---|---|---|---|---|
| None | $87.965 \pm 0.564$ | $68.889 \pm 0.872$ | $83.050 \pm 1.439$ | $39.017 \pm 1.166$ | $70.500 \pm 0.809$ | $75.544 \pm 0.323$ |
| Last layer FA | $\mathbf{89.995} \pm \mathbf{0.647}$ | $69.710 \pm 1.025$ | $80.600 \pm 1.639$ | $48.183 \pm 1.401$ | $70.304 \pm 0.844$ | $75.434 \pm 0.491$ |
| Every layer FA | $56.855 \pm 0.943$ | $71.480 \pm 0.876$ | $83.050 \pm 1.518$ | $\color{red}\mathbf{54.950} \pm 1.331$ | $71.045 \pm 0.909$ | $75.432 \pm 0.475$ |
| DIGL | $74.425 \pm 0.723$ | $63.930 \pm 0.947$ | $81.450 \pm 1.488$ | $37.600 \pm 1.198$ | $71.312 \pm 0.757$ | $54.714 \pm 0.416$ |
| SDRF | $86.825 \pm 0.523$ | $70.210 \pm 0.806$ | $82.700 \pm 1.782$ | $39.583 \pm 1.333$ | $70.696 \pm 0.815$ | $76.480 \pm 0.388$ |
| FoSR | $89.665 \pm 0.416$ | $\color{red}\mathbf{71.810} \pm 0.880$ | $\color{red}\mathbf{86.150} \pm 1.492$ | $45.550 \pm 1.258$ | $\mathbf{74.670} \pm 0.692$ | $\color{red}\mathbf{76.806} \pm 0.451$ |

# 6 CONCLUSIONS

We proposed an efficient graph rewiring method for preventing oversquashing based on iterative first-order maximization of the spectral gap. Further, we identified a shortcoming of existing rewiring approaches that can cause oversmoothing and propose a relational rewiring method to overcome this with theoretical guarantees. Experiments on several graph classification benchmarks demonstrate that the proposed methods can significantly improve the performance of GNNs. In future work it will be interesting to study the effects of oversmoothing and oversquashing in relation to training dynamics and investigate rewiring strategies that aid training.

**Reproducibility Statement**    The computer implementation of the proposed methods along with scripts to re-run our experiments are made publicly available on `https://github.com/kedar2/FoSR`. The experimental settings are described in detail in Section 5 and Appendix D.1, which also details the compute infrastructure and selected hyperparameter values.

ACKNOWLEDGMENTS

This project has been supported by NSF CAREER Grant DMS-2145630, ERC Starting Grant 757983, DFG SPP 2298 (FoDL) Grant 464109215. This work is partly supported by BMBF in DAAD project 57616814 (SECAI).

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

APPENDIX

## A  SPECTRAL GAP AND CHEEGER CONSTANT

### A.1  QUANTIFYING STRUCTURAL BOTTLENECKS VIA THE CHEEGER CONSTANT

Since traditional GNNs use the input graph to propagate neural messages, structural characteristics of the input graph play a crucial role in the quality of signal propagation across nodes. Our method is motivated from the key idea that structural bottlenecks in the input graph lead to information oversquashing in GNNs. A classic measure of "bottlenecked-ness" of a graph is the *Cheeger constant* (Chung and Graham, 1997)

$$h(\mathcal{G}) := \min_{\varnothing \neq \mathcal{S} \subset \mathcal{V}} \frac{|\partial \mathcal{S}|}{\min(\operatorname{vol} \mathcal{S}, \operatorname{vol} \bar{\mathcal{S}})}, \tag{7}$$

where $\operatorname{vol} \mathcal{S} = \sum_{v \in \mathcal{S}} d_v$, $\bar{\mathcal{S}} = \mathcal{V} \setminus \mathcal{S}$ and $\partial \mathcal{S} = \{(u,v) \colon u \in \mathcal{S}, v \in \bar{\mathcal{S}}, (u,v) \in \mathcal{E}\}$ is the edge boundary of $\mathcal{S} \subset \mathcal{V}$. This definition considers the number of edges connecting two complementary subsets of nodes and their volume as measured by the sum of the degrees of the nodes, and then takes the minimum of this value over all possible non-trivial bipartitions of the graph.

Note that $0 \leq h(\mathcal{G}) \leq 1$ for all $\mathcal{S} \subset \mathcal{V}$. $\mathcal{G}$ is connected if and only if $h(\mathcal{G}) > 0$.

Intuitively, when $h(\mathcal{G})$ is large, $\mathcal{G}$ has no structural bottlenecks in the sense that every part of $\mathcal{G}$ is connected to the rest of it by a large fraction of its edges.

A large $h(\mathcal{G})$ implies that there is a low probability of being trapped in a small subset of the nodes and consequently, a simple random walk over the nodes of $\mathcal{G}$ is rapidly mixing (Levin and Peres, 2017).

Well-connected graphs such as the complete graph over $n$ nodes, $K_n$ has a high Cheeger constant, while easily-disconnected graphs have a low Cheeger constant. An example of the latter is the dumbbell graph $K_n$–$K_n$ comprising of two cliques joined by a bridge (see Figure 1), which has a Cheeger constant of $1/(1 + n(n-1))$.

### A.2  BOUNDING THE CHEEGER CONSTANT VIA THE SPECTRAL GAP

In general, computing the exact value of $h(\mathcal{G})$ in practice is known to be a hard problem; see, e.g., Leighton and Rao (1999); Garey et al. (1974); Mohar (1989); Kaibel (2004). Nonetheless, it is possible to bound the Cheeger constant from below and from above in terms of the spectral gap as we explain next. Recall from Section 2.1 that the spectral gap of a graph is the difference $\lambda_2 - \lambda_1$ between the first two eigenvalues of the Laplacian (in increasing order), whereby $\lambda_1 = 0$.

The discrete Cheeger inequality (Cheeger, 1970; Alon and Milman, 1984) shows that the spectral gap of $\mathcal{G}$ provides an estimate of its Cheeger constant:

$$\frac{\lambda_2}{2} \leq h(\mathcal{G}) \leq \sqrt{2\lambda_2}. \tag{8}$$

To intuitively understand this inequality, we can consider the variational formulation of the spectral gap (Chung and Graham, 1997):

$$\lambda_2 = \inf_{x \perp D^{1/2} \mathbf{1}} \frac{x^T L x}{x^T x}. \tag{9}$$

We can write the numerator above as

$$x^T L x = \sum_{(i,j) \in \mathcal{E}} \left( \frac{x_i}{\sqrt{d_i}} - \frac{x_j}{\sqrt{d_j}} \right)^2.$$

Let $\mathcal{S}$ be a subset of the nodes. We may encode $\mathcal{S}$ as a "zero-one vector", writing $x_i = \sqrt{d_i}$ if $i \in \mathcal{S}$ and $x_i = 0$ if $i \notin \mathcal{S}$. This gives us

$$x^T L x = \sum_{(i,j) \in \mathcal{E}} \mathbb{1}_{(i,j) \in \partial \mathcal{S}} = |\partial \mathcal{S}|.$$

That is, we can encode terms from $h(\mathcal{G})$ into the expression for $\lambda_2$ if we use this variational formulation.

The discrete Cheeger inequality tells us that the spectral gap is bounded away from zero if and only if $h(\mathcal{G})$ is bounded away from zero. $\mathcal{G}$ is connected if and only if $\lambda_2 > 0$. Since the spectral gap can be computed efficiently, we use it as a measure of graph bottlenecked-ness in lieu of the Cheeger constant.

## B    SPECTRAL GAP AND OVERSQUASHING

In this section, we provide some more intuition behind our idea of optimizing the spectral gap of the input graph for alleviating structural bottlenecks and improving the overall quality of information propagation across nodes.

The *rate of spectral expansion*, i.e., the rate at which the spectral gap improves as a function of the rewiring iterations is one of the key determinants of oversquashing (Banerjee et al., 2022), and a faster rate is one of the main features of our new algorithm. We show that FoSR has a faster rate of spectral expansion compared to existing curvature-based methods such as SDRF (Topping et al., 2022) for both the synthetic benchmark NEIGHBORSMATCH (Alon and Yahav, 2021) (Appendix B.1.1) and the TUDataset (Morris et al., 2020) (Appendix B.1.2).

In Appendix B.2, we show that optimizing only the first-order change in the spectral gap gives FoSR a significant computational advantage over SDRF. In Appendix B.3, we empirically explore the trade-off between oversquashing and oversmoothing as a function of the number of rewiring iterations. Finally in Appendix B.4, we discuss the approximation error of FoSR.

### B.1    COMPARING THE RATE OF SPECTRAL EXPANSION FOR FOSR AND SDRF

#### B.1.1    THE NEIGHBORSMATCH PROBLEM (DETAILS ON FIGURE 1)

Alon and Yahav (2021) introduced the synthetic benchmark NEIGHBORSMATCH to test the impact of oversquashing in GNNs. Given an input graph $\mathcal{G}$, suppose that we wish to predict the label for a node T; see Figure 2(a). The correct label is the label of the green node that has the same number of blue neighbors as T. For the example in Figure 2(a), the answer is B which happens to reside at the opposite end of the graph. A correct prediction of the label for this example will require a number of GNN layers $l$ that is equal to the diameter of the graph. With increasing $l$, however, at each message-passing layer, information from exponentially-growing receptive fields need to be concurrently propagated. This leads to oversquashing and the GNN fails to propagate long-range signals and fit the training dataset perfectly.

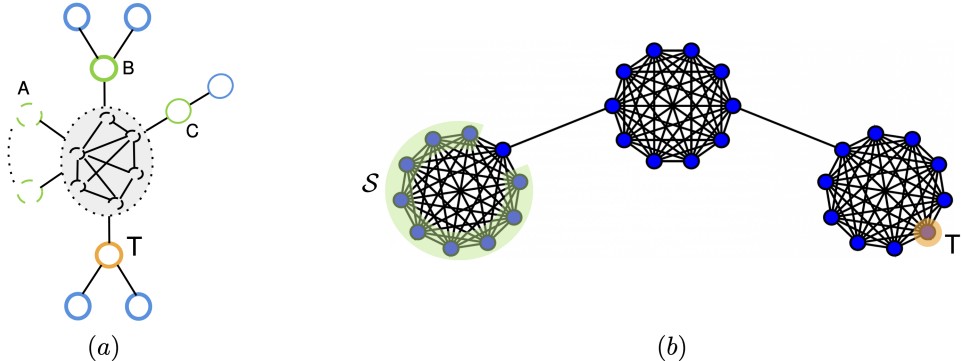

$(a)$ $\qquad\qquad\qquad\qquad\qquad\qquad\qquad\qquad$ $(b)$

Figure 2: Learning task modeled on the NEIGHBORSMATCH problem.

We consider a learning task modeled on the NEIGHBORSMATCH problem. Given an input graph $\mathcal{G}$, a target node T, and a subset $\mathcal{S}$ of the nodes of $\mathcal{G}$, we assign each node in $\mathcal{S}$ a different random $|\mathcal{S}|$-dimensional one-hot vector encoding the number of blue neighbors. Likewise, we represent the label of T as a random $|\mathcal{S}|$-dimensional one-hot vector and the goal is to predict the node T' $\in \mathcal{S}$ with

the same label as T. We take the input $\mathcal{G}$ to be a path-of-cliques, which comprises of three cliques each of size 10 connected in a path by two edges; see Figure 2(b). The set $\mathcal{S}$ consists of the first 9 nodes in clique-1, and the target T is the last node of clique-3. The range of the interaction, i.e., the maximum distance between T and a node in $\mathcal{S}$ is 5. We trained a graph attention network (Veličković et al., 2018) with 6 layers each of width 64 on a training dataset comprising of 10000 copies of $G$, where each copy has a different mapping matching the target.

Figure 1 (bottom) shows the evolution of the normalized spectral gap and training accuracy as a function of the number of rewiring iterations for three algorithms: our FoSR, SDRF (Topping et al., 2022), and G-RLEF (Banerjee et al., 2022). SDRF is motivated from the idea that negatively curved edges lead to oversquashing. The notion of curvature, which can be construed as a "local" Cheeger constant, plays a key role in the SDRF rewiring process. SDRF cycles between adding a supporting edge around a negatively curved edge and removing a redundant positively-curved edge; see Figure 1 (top). Since the addition and removal of edges are done independently of each other, SDRF can potentially disconnect a connected input graph. G-RLEF, on the other hand, seeks to improve the "global" Cheeger constant $h(\mathcal{G})$ sampling edges according to inverse triangle counts (a proxy for curvature) and using a local edge flip mechanism so that the input graph is never disconnected. In our experiments, we configured SDRF to only add edges. For all three algorithms, we observe that both the normalized spectral gap and training accuracy increase monotonically in the number of rewiring steps until they saturate at around 150 iterations. The rate of spectral expansion i.e., the rate at which the spectral gap improves as a function of the rewiring iterations is the fastest for FoSR.

### B.1.2 THE TUDATASET

Figure 3 shows the normalized spectral gap as a function of the number of rewiring iterations for FoSR and SDRF on the TUDataset (Morris et al., 2020). Again, we observe that FoSR has a much faster rate of spectral expansion compared to that of the SDRF (Topping et al., 2022).

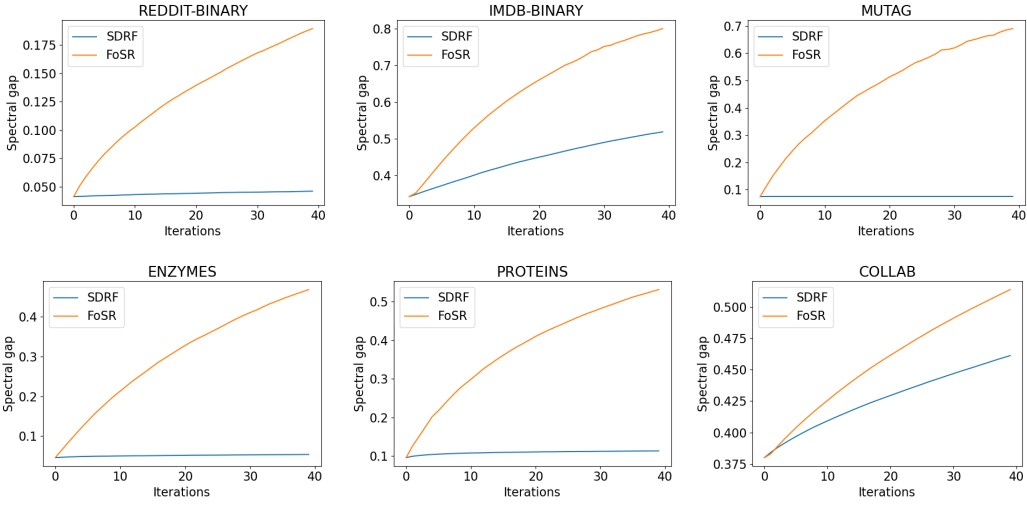

Figure 3: Normalized spectral gap as a function of the number of rewiring iterations for FoSR and SDRF on the TUDataset graphs (Morris et al., 2020). We record the average spectral gap across all graphs in the dataset for each iteration count.

### B.2 COMPARISON OF THE COMPUTATION TIME FOR FoSR AND SDRF

Our First-order Spectral Rewiring (FoSR) method computes the first-order change in the spectral gap from adding each edge, and then adds the edge which maximizes this. Optimizing only the first-order change gives our method a significant computational advantage over competing curvature-based methods like the SDRF (Topping et al., 2022). Figure 4 and Table 2 show the computation time of FoSR and SDRF for a synthetic dataset consisting of Erdös-Rényi graphs, and the TUDataset graphs

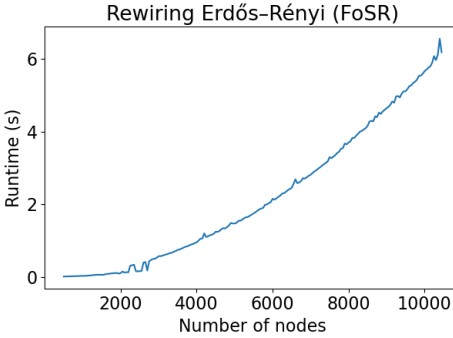 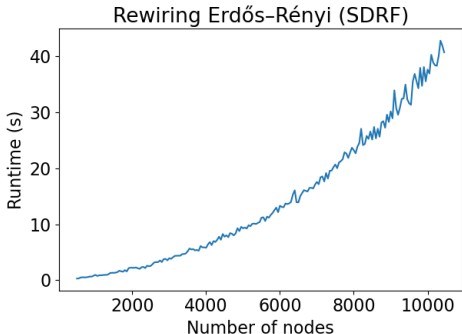

Figure 4: Compute time of FoSR and SDRF for Erdös-Rényi graphs. For a graph with $n$ nodes, we add an edge $(i, j)$ with probability $p = \frac{5 \log n}{n}$.

(Morris et al., 2020) respectively. We observe that the run times for FoSR are almost an order of magnitude lower than that of SDRF.

Table 2: Runtime to rewire every graph for 10 iterations (in seconds).

| Architecture | REDDIT-BINARY | IMDB-BINARY | MUTAG | ENZYMES | PROTEINS | COLLAB |
|---|---|---|---|---|---|---|
| FoSR | 37.4167 | 0.0924058 | 0.0146499 | 0.0622702 | 0.1677542 | 4.58400 |
| SDRF | 193.432 | 5.28793 | 0.407810 | 2.38515 | 5.51248 | 610.542 |

### B.3 TRADE-OFF BETWEEN OVERSQUASHING AND OVERSMOOTHING AS A FUNCTION OF REWIRING ITERATIONS

Table 3 shows that the relational architecture indeed helps to reduce oversmoothing, since the relational rewiring leads to higher Dirichlet energies. Here the Dirichlet energy is measured as an average over all graphs in the dataset after training, summed over 100 runs.

Relational GNNs evidently help to reduce oversmoothing. However, the tradeoff between oversquashing and oversmoothing still exists. Recall that input graphs with small spectral gaps are prone to oversquashing, whereas inputs with large spectral gaps are prone to oversmoothing (see Section 3.2). We use R-GNNs to alleviate oversmoothing, but there is still an optimal number of edges to add, beyond which performance declines. Figure 5 shows that the performance for R-GCNs peaks at around 25 iterations, which is reflected in both the test accuracy and Dirichlet energy of the final layer.

### B.4 DISCUSSION ON THE APPROXIMATION ERROR OF FOSR

When rewiring a graph using FoSR, the objective at each iteration is to add the edge which maximally increases the spectral gap. Since FoSR makes this calculation using a first-order approximation of the spectral gap, there is some error incurred between the estimated spectral gap and the actual spectral gap. This error is caused by two factors. The first is the error from using the first-order change of the spectral gap instead of the actual change. The second error accrues from using the dominant term of the first-order change instead of the first-order change. We record each of these three quantities (actual change, first-order change, FoSR approximated change) for graphs in the ENZYMES dataset (Morris et al., 2020) in Figure 6. We see that the actual change is well-correlated with the approximations, and that the approximations are well-correlated with each other.

While FoSR accumulates some error from the actual change in the spectral gap, we are primarily interested in how this affects its ability to choose good edges for rewiring. Hence, it is natural to ask how FoSR compares to a greedy algorithm which computes the spectral gap exactly. We test the

Table 3: Dirichlet energy of the final layer for TUDataset graphs after training. The relational architectures tend to provide higher energy, which indicates that they are less prone to oversmoothing.

**GCN**

| Rewiring | REDDIT-BINARY | IMDB-BINARY | MUTAG | ENZYMES | PROTEINS | COLLAB |
|---|---|---|---|---|---|---|
| None | $0.6188 \pm 0.0676$ | $0.0018 \pm 0.0001$ | $0.0522 \pm 0.0046$ | $0.0954 \pm 0.0026$ | $0.0470 \pm 0.0008$ | $0.0026 \pm 0.0001$ |
| DIGL | $0.0002 \pm 0.0000$ | $0.0005 \pm 0.0001$ | $0.0303 \pm 0.0032$ | $0.0392 \pm 0.0007$ | $0.0398 \pm 0.0010$ | $0.0001 \pm 0.0000$ |
| SDRF | $0.6410 \pm 0.0584$ | $0.0013 \pm 0.0001$ | $0.0504 \pm 0.0047$ | $0.0927 \pm 0.0023$ | $0.0468 \pm 0.0007$ | $0.0025 \pm 0.0001$ |
| FoSR | $0.6047 \pm 0.0497$ | $0.0020 \pm 0.0001$ | $0.0579 \pm 0.0024$ | $0.0844 \pm 0.0023$ | $0.0699 \pm 0.0013$ | $0.0030 \pm 0.0005$ |

**R-GCN**

| Rewiring | REDDIT-BINARY | IMDB-BINARY | MUTAG | ENZYMES | PROTEINS | COLLAB |
|---|---|---|---|---|---|---|
| None | $0.8077 \pm 0.0007$ | $0.0607 \pm 0.0001$ | $0.0966 \pm 0.0040$ | $0.1214 \pm 0.0024$ | $0.1161 \pm 0.0043$ | $0.0904 \pm 0.0001$ |
| DIGL | $0.0425 \pm 0.0000$ | $0.0309 \pm 0.0000$ | $0.0918 \pm 0.0046$ | $0.0894 \pm 0.0015$ | $0.0979 \pm 0.0045$ | $0.0041 \pm 0.0000$ |
| SDRF | $1.8460 \pm 0.0164$ | $0.0613 \pm 0.0061$ | $0.1575 \pm 0.0101$ | $0.2411 \pm 0.0069$ | $0.1235 \pm 0.0047$ | $0.8306 \pm 0.0475$ |
| FoSR | $1.3733 \pm 0.0203$ | $1.3040 \pm 0.0705$ | $0.5345 \pm 0.0201$ | $0.3206 \pm 0.0088$ | $0.4308 \pm 0.0337$ | $0.8139 \pm 0.0382$ |

**GIN**

| Rewiring | REDDIT-BINARY | IMDB-BINARY | MUTAG | ENZYMES | PROTEINS | COLLAB |
|---|---|---|---|---|---|---|
| None | $1.1215 \pm 0.0379$ | $0.0816 \pm 0.0093$ | $0.1287 \pm 0.0053$ | $0.1064 \pm 0.0018$ | $0.0853 \pm 0.0034$ | $0.1128 \pm 0.0074$ |
| DIGL | $0.1080 \pm 0.0103$ | $0.0517 \pm 0.0061$ | $0.0442 \pm 0.0052$ | $0.0675 \pm 0.0011$ | $0.0679 \pm 0.0029$ | $0.0067 \pm 0.0011$ |
| SDRF | $1.0996 \pm 0.0392$ | $0.0694 \pm 0.0092$ | $0.1126 \pm 0.0047$ | $0.1005 \pm 0.0015$ | $0.0783 \pm 0.0027$ | $0.1164 \pm 0.0075$ |
| FoSR | $0.8914 \pm 0.0522$ | $0.0916 \pm 0.0148$ | $0.1314 \pm 0.0068$ | $0.1298 \pm 0.0027$ | $0.0781 \pm 0.0058$ | $0.1271 \pm 0.0098$ |

**R-GIN**

| Rewiring | REDDIT-BINARY | IMDB-BINARY | MUTAG | ENZYMES | PROTEINS | COLLAB |
|---|---|---|---|---|---|---|
| None | $1.2972 \pm 0.0336$ | $0.2625 \pm 0.0235$ | $0.4140 \pm 0.0142$ | $0.1825 \pm 0.0039$ | $0.3021 \pm 0.0286$ | $0.3293 \pm 0.0244$ |
| DIGL | $0.3265 \pm 0.0271$ | $0.0591 \pm 0.0075$ | $0.0772 \pm 0.0052$ | $0.1319 \pm 0.0021$ | $0.2534 \pm 0.0168$ | $0.0175 \pm 0.0016$ |
| SDRF | $1.5525 \pm 0.0260$ | $0.1031 \pm 0.0051$ | $0.2754 \pm 0.0102$ | $0.1768 \pm 0.0027$ | $0.2373 \pm 0.0144$ | $0.2695 \pm 0.0153$ |
| FoSR | $0.8037 \pm 0.0227$ | $0.1828 \pm 0.0125$ | $0.2901 \pm 0.0063$ | $0.1000 \pm 0.0010$ | $0.1593 \pm 0.0051$ | $0.2414 \pm 0.0129$ |

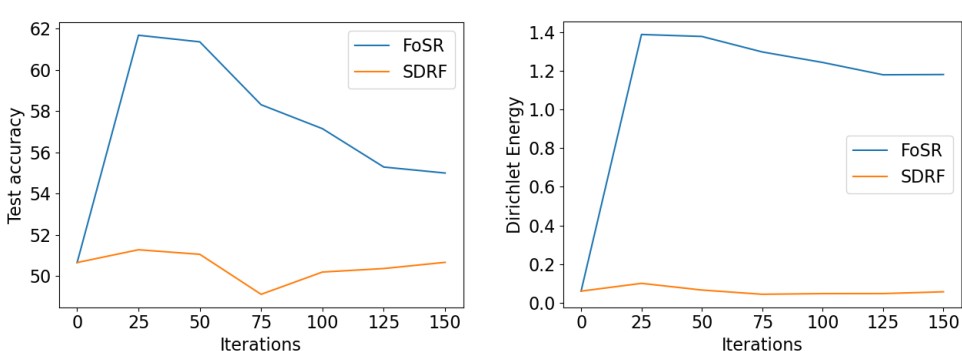

Figure 5: Trade-off between oversquashing and oversmoothing as a function of rewiring iterations for the R-GCN on the IMDB-BINARY graph classification task. Left: Test accuracy as a function of the rewiring iterations. Right: Dirichlet energy of the final layer as a function of rewiring iterations. The performance for R-GCNs peaks at around 25 iterations, which is reflected in both the test accuracy and Dirichlet energy.

performance of FoSR on a synthetic dumbbell graph, consisting of two cliques of size 50, connected via a path of length 3. The results are shown in Figure 7. We see that the spectral evolution of FoSR is indistinguishable from that of the greedy method, indicating that the approximation is strong and that the two methods choose similar edges.

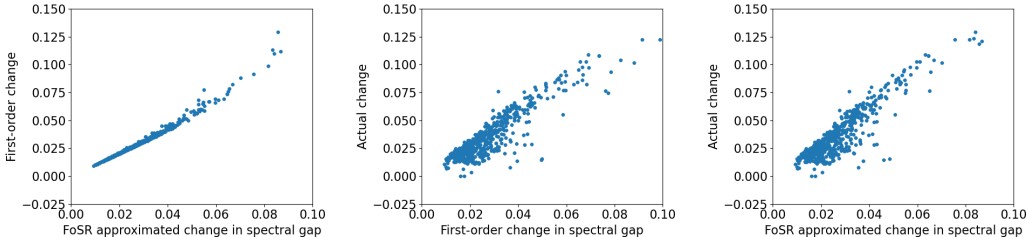

Figure 6: Change in the spectral gap, first-order change, and the FoSR approximated change for rewired graphs. The changes are recorded for each graph in the ENZYMES dataset after 1 rewiring iteration.

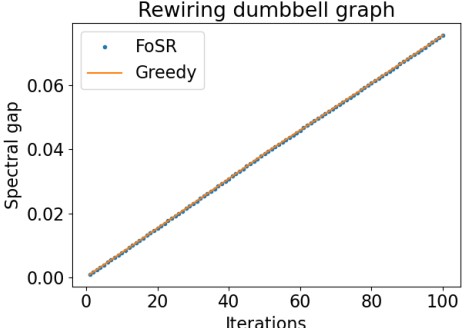

Figure 7: Evolution of the spectral gap of a dumbbell graph under rewiring, evaluated at every iteration count between 0 and 100. The performance of FoSR is nearly identical to a method which selects edges to add via exact computation of eigenvalues.

## C   Details on Section 4

*Proof of Theorem 4.* We refer to Stewart and Sun (1990, Theorem 2.3) for a proof that $\lambda_i$ is a continuously differentiable function and a computation of its gradient for general classes of matrices. We provide a calculation for our case here.

Fix a symmetric matrix $M_0 \in \mathbb{R}^{n \times n}$ and let $M_0 = U\Sigma U^{-1}$ be an orthonormal diagonalization of $M_0$ with the diagonal entries of $\Sigma$ sorted in descending order. Fix $j, k \in [n]$. Let $E_{j,k}$ denote the matrix whose $(j, k)$ entry is 1 and remaining entries are 0. Define a curve $\gamma : \mathbb{R} \to \mathbb{R}^{n \times n}$ by

$$\gamma(t) := M_0 + tUE_{j,k}U^{-1}.$$

Then

$$\lambda_i(\gamma(t)) = \lambda_i(U(\Sigma + tE_{j,k})U^{-1}) = \lambda_i(\Sigma + tE_{j,k}) = \lambda_i + \delta_{i,j}\delta_{i,k}t, \tag{10}$$

where $\delta_{i,j}$ is equal to 1 if $i = j$ and 0 otherwise. The final equality follows since the $i$-th diagonal entry of $\Sigma$ is $\lambda_i$, and adding an off-diagonal entry does not change its eigenvalues. For sufficiently small $t$ the order of the eigenvalues does not change. By the chain rule,

$$\frac{d}{dt}\Big|_{t=0} \lambda_i(\gamma(t)) = (d\lambda_i)_{M_0}(\dot{\gamma}(0)).$$

By equation (10), the left-hand side of the above equation is equal to $\delta_{i,j}\delta_{i,k}$. So

$$\delta_{i,j}\delta_{i,k} = (d\lambda_i)_{M_0}(\dot{\gamma}(0)) = (d\lambda_i)_{M_0}(UE_{j,k}U^{-1}).$$

By the definition of the gradient,

$$\mathrm{Tr}(UE_{j,k}U^{-1}\nabla\lambda_i(M_0)) = (d\lambda_i)_{M_0}(UE_{j,k}U^{-1}) = \delta_{i,j}\delta_{i,k} = \mathrm{Tr}(E_{i,i}E_{j,k}).$$

Rewriting the left-hand side of the above equation gives us

$$\mathrm{Tr}(U^{-1}\nabla\lambda_i(M_0)UE_{j,k}) = \mathrm{Tr}(E_{i,i}E_{j,k}).$$

Since this holds for all $j, k \in [n]$ and the $E_{j,k}$ span $\mathbb{R}^{n \times n}$, this implies that

$$U^{-1}\nabla\lambda_i(M_0)U = E_{i,i}$$
$$\nabla\lambda_i(M_0) = UE_{i,i}U^{-1}$$
$$\nabla\lambda_i(M_0) = x_i x_i^T,$$

as desired. □

*Proof of Proposition 5.* Let $A$ be the adjacency matrix of a graph $G$, and let $A_N$ be the normalized adjacency matrix. Let $x$ be an eigenvector of $A_N$ with eigenvalue $\lambda$. Suppose that we add an edge $(u, v)$ to $G$. Let $\delta A_N$ denote the entry-wise change in the normalized adjacency matrix from adding the edge $(u, v)$. If $i \neq u$ and $j \neq v$, then

$$(\delta A_N)_{ij} = 0.$$

If $i \neq u$ and $j = v$, then

$$(\delta A_N)_{ij} = \frac{A_{ij}}{\sqrt{d_i}}\left(\frac{1}{\sqrt{1 + d_j}} - \frac{1}{\sqrt{d_j}}\right).$$

If $i = u$ and $j \neq v$, then

$$(\delta A_N)_{ij} = \frac{A_{ij}}{\sqrt{d_j}}\left(\frac{1}{\sqrt{1 + d_i}} - \frac{1}{\sqrt{d_i}}\right).$$

If $i = u$ and $j = v$, then

$$(\delta A_N)_{ij} = \frac{1}{(1 + \sqrt{d_i})(1 + \sqrt{d_j})}.$$

By Theorem 4, the first-order change in $\lambda_2(A_N)$ is given by

$$x^T(\delta A_N)x = \sum_{i=u}\sum_{j=v}(\delta A_N)_{ij}x_ix_j + \sum_{i=u}\sum_{j\neq v}(\delta A_N)_{ij}x_ix_j + \sum_{i=v}\sum_{j=u}(\delta A_N)_{ij}x_ix_j + \sum_{i=v}\sum_{j\neq u}(\delta A_N)_{ij}x_ix_j$$

$$+ \sum_{i\neq u,v}\sum_{j=u}(\delta A_N)_{ij}x_ix_j + \sum_{i\neq u,v}\sum_{j=v}(\delta A_N)_{ij}x_ix_j + \sum_{i\neq u,v}\sum_{j\neq u,v}(\delta A_N)_{ij}x_ix_j$$

$$= \frac{x_ux_v}{(\sqrt{1+d_u})(\sqrt{1+d_v})} + \sum_{j\neq v}\frac{A_{uj}x_ux_j}{\sqrt{d_j}}\left(\frac{1}{\sqrt{1+d_u}} - \frac{1}{\sqrt{d_u}}\right)$$

$$+ \frac{x_ux_v}{(\sqrt{1+d_u})(\sqrt{1+d_v})} + \sum_{j\neq u}\frac{A_{vj}x_vx_j}{\sqrt{d_j}}\left(\frac{1}{\sqrt{1+d_v}} - \frac{1}{\sqrt{d_v}}\right)$$

$$+ \sum_{i\neq u,v}\frac{A_{iu}x_ix_u}{\sqrt{d_i}}\left(\frac{1}{\sqrt{1+d_u}} - \frac{1}{\sqrt{d_u}}\right) + \sum_{i\neq u,v}\frac{A_{iv}x_ix_v}{\sqrt{d_i}}\left(\frac{1}{\sqrt{1+d_v}} - \frac{1}{\sqrt{d_v}}\right)$$

$$= \frac{2x_ux_v}{(\sqrt{1+d_u})(\sqrt{1+d_v})} + 2\sum_{i\neq u}\frac{A_{iv}x_ix_v}{\sqrt{d_i}}\left(\frac{1}{\sqrt{1+d_v}} - \frac{1}{\sqrt{d_v}}\right)$$

$$+ 2\sum_{i\neq v}\frac{A_{iu}x_ix_u}{\sqrt{d_i}}\left(\frac{1}{\sqrt{1+d_u}} - \frac{1}{\sqrt{d_u}}\right)$$

$$= \frac{2x_ux_v}{(\sqrt{1+d_u})(\sqrt{1+d_v})} + 2\sum_{i\neq u}(A_N)_{iv}x_ix_v\left(\frac{\sqrt{d_v}}{\sqrt{1+d_v}} - 1\right)$$

$$+ 2\sum_{i\neq v}(A_N)_{iu}x_ix_u\left(\frac{\sqrt{d_u}}{\sqrt{1+d_u}} - 1\right)$$

$$= \frac{2x_ux_v}{(\sqrt{1+d_u})(\sqrt{1+d_v})} + 2\sum_{i}(A_N)_{iv}x_ix_v\left(\frac{\sqrt{d_v}}{\sqrt{1+d_v}} - 1\right)$$

$$+ 2\sum_{i}(A_N)_{iu}x_ix_u\left(\frac{\sqrt{d_u}}{\sqrt{1+d_u}} - 1\right)$$

Since $x$ is an eigenvector of $A_N$ with eigenvalue $\lambda$, we may write the above expression as

$$\frac{2x_ux_v}{(\sqrt{1+d_u})(\sqrt{1+d_v})} + 2\lambda x_v^2\left(\frac{\sqrt{d_v}}{\sqrt{1+d_v}} - 1\right) + 2\lambda x_u^2\left(\frac{\sqrt{d_u}}{\sqrt{1+d_u}} - 1\right).$$

Applying this equality to the second eigenvector of $A_N$ yields the result. $\qquad\square$

## D   DETAILS ON THE EXPERIMENTS FROM SECTION 5

The computer implementation of the proposed methods along with scripts to re-run our experiments are made publicly available on `https://anonymous.4open.science/r/FoSR-0CE3/`. In the following we list details on hyperparameters, libraries, and the compute infrastructure we used in these experiments.

### D.1   HYPERPARAMETERS

Table 4: Common hyperparameters

| | |
|---|---|
| Dropout | 0.5 |
| Number of layers | 4 |
| Hidden dimension | 64 |
| Learning rate | $10^{-3}$ |
| Stopping patience | 100 epochs |

Table 5: Rewiring iteration counts for FoSR and SDRF.

FoSR

| Architecture | REDDIT-BINARY | IMDB-BINARY | MUTAG | ENZYMES | PROTEINS | COLLAB |
|---|---|---|---|---|---|---|
| GCN | 5 | 5 | 40 | 10 | 20 | 10 |
| R-GCN | 5 | 20 | 40 | 40 | 5 | 5 |
| GIN | 10 | 20 | 20 | 5 | 10 | 20 |
| R-GIN | 40 | 20 | 5 | 40 | 20 | 10 |

SDRF

| Architecture | REDDIT-BINARY | IMDB-BINARY | MUTAG | ENZYMES | PROTEINS | COLLAB |
|---|---|---|---|---|---|---|
| GCN | 5 | 20 | 5 | 5 | 40 | 5 |
| R-GCN | 40 | 5 | 40 | 5 | 20 | 20 |
| GIN | 5 | 10 | 5 | 5 | 20 | 40 |
| R-GIN | 5 | 40 | 5 | 5 | 5 | 20 |

Table 6: DIGL hyperparameters.

Teleport probability ($\alpha$)

| Architecture | REDDIT-BINARY | IMDB-BINARY | MUTAG | ENZYMES | PROTEINS | COLLAB |
|---|---|---|---|---|---|---|
| GCN | 0.15 | 0.05 | 0.15 | 0.15 | 0.15 | 0.05 |
| R-GCN | 0.15 | 0.05 | 0.05 | 0.15 | 0.05 | 0.05 |
| GIN | 0.05 | 0.15 | 0.05 | 0.15 | 0.15 | 0.15 |
| R-GIN | 0.05 | 0.15 | 0.05 | 0.05 | 0.05 | 0.05 |

Sparsification threshold ($\epsilon$)

| Architecture | REDDIT-BINARY | IMDB-BINARY | MUTAG | ENZYMES | PROTEINS | COLLAB |
|---|---|---|---|---|---|---|
| GCN | $10^{-3}$ | $10^{-4}$ | $10^{-3}$ | $10^{-4}$ | $10^{-4}$ | $10^{-4}$ |
| R-GCN | $10^{-4}$ | $10^{-3}$ | $10^{-3}$ | $10^{-3}$ | $10^{-3}$ | $10^{-3}$ |
| GIN | $10^{-3}$ | $10^{-4}$ | $10^{-4}$ | $10^{-3}$ | $10^{-3}$ | $10^{-4}$ |
| R-GIN | $10^{-3}$ | $10^{-4}$ | $10^{-3}$ | $10^{-4}$ | $10^{-4}$ | $10^{-3}$ |

## D.2 COMPUTER INFRASTRUCTURE AND LIBRARIES

In this section, we provide details of our implementation. All the experiments were implemented in Python using PyTorch (Paszke et al., 2019), NumPy (Harris et al., 2020), PyG (PyTorch Geometric) (Fey and Lenssen, 2019), with plots created using Matplotlib (Hunter, 2007). PyTorch, PyG and NumPy are made available under the BSD license, and Matplotlib under the PSF license.

We conducted all our experiments on a local server with 2x 22-Core/44-Thread Intel Xeon Scalable Gold 6152 processor (2.1/3.7 GHz) and 8x NVIDIA GeForce RTX 2080 Ti graphics card (11 GB, GDDR6).

