# OpenReview forum: "FoSR: First-order spectral rewiring for addressing oversquashing in GNNs"
_ICLR.cc/2023/Conference — ICLR 2023 poster_

### Official Review · Reviewer_T9j4 · 2022-10-19

**Confidence:** 4
**Correctness:** 3
**Technical Novelty And Significance:** 3
**Empirical Novelty And Significance:** 3
**Recommendation:** 6

**Clarity, Quality, Novelty And Reproducibility:**

- Clarity: the paper is well-written and easy to understand the main idea of the method.

- Quality: I have some concerns about the motivation and the experiments. Please see my comments above.

- Novelty: The proposed method seems novel to me. I like the idea of leveraging relational GNN especially.

- Reproducibility: The code is complemented with a well-documented readme file. I believe the results are reproducible, albeit I do not reproduce them directly.


**Strength And Weaknesses:**

## Strengths

- Proposing the use of relational GNN to preserve the original graph topological information is reasonable.

- The proposed edge-adding method, FoSR (First-order Spectral Rewiring), can maximize the spectral gap in a greedy but effective way.

- The experimental results on the graph classification task demonstrate the superior performance of the proposed method in terms of test accuracy.

- The experiment setting is rigorous. The code seems to be well-documented.

## Weaknesses

- The criterion of maximizing the spectral gap for adding edges seems not well-motivated.

- The experiment results do not support or validate the theoretical analysis and claims. Also, the efficiency (i.e. time complexity) of FoSR is not reported and compared with baseline methods.

- Some naive baselines (i.e. fully connected graph at the last layer) and transformer-based methods are only mentioned but not included in the experiments.

## Detail comments

I have mixed feelings about this work. On one hand, I like the idea of leveraging relational GNNs to preserve the original graph information is great for graph rewiring-based methods. I also love how rigorous the authors prepared their experiments and make their code well-documented (at least specifying the environment).

On the other hand, I have several concerns about the paper. I explain all the listed weaknesses below.

### The criterion of maximizing the spectral gap for adding edges seems not well-motivated.

As the authors mentioned in the Section 3, once we introduce relational GNNs, we will not suffer from losing the original topological information (i.e. $\mathcal{E}_1$). As also mentioned by the authors, using a complete graph can maximally resolve the oversquashing issue, albeit it comes with the cost of losing original topological information. However, this is not a problem when we consider relational GNNs. Hence, should not we simply choose $\mathcal{E}_2 = \mathcal{E}/\mathcal{E}_1$? I do not get the motivation why we need a special design of $\mathcal{E}_2$ for solving the oversquashing problem theoretically.

Note that the naive approach abovementioned has an obvious drawback, which is its scalability on large graphs similar to the transformer-based method. However, given the fact that the authors only consider graph classification tasks, I feel like there is no need for the proposed FoSR method. Maybe I miss something, please correct me if I do. If not, I would suggest the authors test these rewiring methods on node classification tasks, where the graph can be huge.

### The experiment results do not fully support or validate the theoretical analysis and claims. Also, the efficiency (i.e. time complexity) of FoSR is not reported and compared with baseline methods.

I notice that the authors demonstrate that their method indeed achieves a higher spectral gap compared to the baselines in Figure 1. I wonder if the authors also report similar results for their main experiments? Note that this is quite important, as the authors claim in the introduction that the spectral gap is a measure of the oversquashing effect.

Also, one major contribution of the paper is their first-order approximation formalism, which makes the FoSR algorithm much faster in theory. However, I wonder how the running time of FoSR compared to the other baselines. Does FoSR win in both test accuracy and time complexity? If not, what does the trade-off look like?

### Some naïve baselines (i.e. fully connected graph at the last layer) and transformer-based methods are only mentioned but not included in the experiments.

One major drawback of the experiments is the selection of baseline methods. As mentioned by the authors in the introduction, some naive methods can resolve the oversquashing problem. For example, leveraging a complete graph in the last layer of the GNN. The other naive baseline would be the one that I proposed earlier, which simply chooses $\mathcal{E}_2 = \mathcal{E}/\mathcal{E}_1$ for relational GNN. Finally, note that the transformer-based methods will not suffer from oversquashing and perform pretty well on graph classification tasks recently. I wonder how the proposed methods compare to transformer-based methods. Do we still need these rewiring techniques for graph classification? As I mentioned earlier, if the authors want to prevent comparison with transformer-based methods or methods leveraging complete graphs, they should conduct experiments on node classification tasks with large graphs.




**Summary Of The Paper:**

The authors propose a graph rewiring method, which aims to tackle the oversquashing problem in GNN literature. The proposed method greedily adds edges for maximizing the spectral gap of the graph sequentially. The authors also propose the use of relational GNNs to preserve the original graph topological information. The experimental results show that the proposed method achieves better test accuracy on graph classification tasks compared to several recently proposed rewiring methods.

**Summary Of The Review:**

The authors attempt to tackle the oversquashing problem and propose a new rewiring method and the use of relational GNN. Their proposed method aims to add edges that maximize the spectral gap. My major concern about the paper is its motivation, where the complete graph can simply resolve the oversquashing problem and the original graph information can be preserved by the use of relational GNN. The experimental results only demonstrate the better test accuracy of the proposed method on graph classification tasks, where some simple baseline and transformer-based methods are ignored. Also, the comparison of the time efficiency is not presented. Hence, I believe the paper needs some revision before publishing.

====================post rebuttal===========================

I thank the response by the authors. Most of my concerns are well-addressed but one still remains (please see my response below). Nevertheless, based on the merits of the current paper, I raise my rating from 5 to 6.

---

> ### Author Response · Authors · 2022-11-17
> **Initial response to Reviewer T9j4 (part 1)**
>
> Thanks for your valuable feedback and suggestions. We have made systematic efforts to address your comments and suggestions as detailed below.
>
>
> > The criterion of maximizing the spectral gap for adding edges seems not well-motivated.
>
> Thanks for the comment. First we observe that the method works, as can be seen in the results reported in Table 1 in Section 5. We have expanded the theoretical motivation in Appendix A, with additional experiments in Appendix B, which we describe in the following.
>
> - For the motivation, we have added details to the theoretical discussion in Appendix A, where we explain why the spectral gap is a well known and useful quantity to measure structural bottlenecks in an input graph and hence is a good candidate for measuring oversquashing.
>
>  - We have provided evidence that the spectral gap improves connectivity, which can help improve test accuracy initially. At the same time, we observe that the test accuracy does not monotonically increase with the spectral gap because at some point oversmoothing starts to become more pronounced. We have empirically demonstrated this tradeoff between oversquashing and oversmoothing (as a function of rewiring iterations, and hence the spectral gap) in Figure 5. For a detailed discussion, please see the new Appendix B.3.
>
> - We also included several additional experiments we report in Appendix B. Please see our "Common response: Summary of main updates to the manuscript".
>
> > As the authors mentioned in the Section 3, once we introduce relational GNNs, we will not suffer from losing the original topological information (i.e. $E_1$). As also mentioned by the authors, using a complete graph can maximally resolve the oversquashing issue, albeit it comes with the cost of losing original topological information. However, this is not a problem when we consider relational GNNs. Hence, should not we simply choose $\mathcal{E}_2 = \mathcal{E}/\mathcal{E}_1$? I do not get the motivation why we need a special design of $\mathcal{E}_2$ for solving the oversquashing problem theoretically.
>
>
> This a good point. It is correct that the fully connected network will have good (in the fact the best) connectivity. We also observe this in Figure 1 and the new Figure 3, where the connectivity, as measured by the spectral gap, increases with the number of added edges. The relational architecture is indeed more robust to oversmoothing than its non-relational counterpart, so this is a better strategy for the relational case than the non-relational case. We have added a clear comparison demonstrating this in the new Table 3. At the same time, even for the relational case, we observe that there is an optimal number of edges to add in terms of test accuracy beyond which the Dirichlet energy of the final layer decreases, resulting in some amount of oversmoothing. To demonstrate this we have added the new Figure 5 showing the tradeoff between number of added edges and test accuracy and Dirichlet energy (an indicator of oversmoothing). The conclusion is that if we include too many edges in rewiring, the training accuracy will suffer compared to adding a moderate number of edges. Hence our strategy to selectively add edges is more adequate than indiscriminately adding edges or considering a fully connected graph.
>
> To give an intuitive explanation for these experimental observations:  Suppose that we weight the new edges heavily in the relational structure relative to the existing edges. Then the GNN will forget the original graph structure and assign similar representations to every node, resulting in oversmoothing. If on the other hand we decide to weight the new edges very weakly, then we will be back in a situation similar to the original graph without any rewiring.
>
> This is consistent with our Theorem 3, since the relational GNN with added edges still performs better than the original GNN without rewiring.
>
> This shows in particular that **fully adjacent graphs do not work better even if one uses a relational structure**.
>
> We hope this addresses your concerns.
>
> > I notice that the authors demonstrate that their method indeed achieves a higher spectral gap compared to the baselines in Figure 1. I wonder if the authors also report similar results for their main experiments? Note that this is quite important, as the authors claim in the introduction that the spectral gap is a measure of the oversquashing effect.
>
> Thanks for your comment. We have now added figures for our main experiments: Figure 3 (in Appendix B.1.2) shows the normalized spectral gap as a function of the number of rewiring iterations for the TUDataset graphs. The figures show that **FoSR indeed achieves a fast rate of spectral expansion on real world data sets**.

---

> > ### Author Response · Authors · 2022-11-17
> > **Initial response to Reviewer T9j4 (part 2)**
> >
> > > Also, one major contribution of the paper is their first-order approximation formalism, which makes the FoSR algorithm much faster in theory. However, I wonder how the running time of FoSR compared to the other baselines. Does FoSR win in both test accuracy and time complexity? If not, what does the tradeoff look like?
> >
> > Thanks for your comment. We have added Appendix B.2, where we show that optimizing only the first-order change in the spectral gap gives FoSR a significant computational advantage over SDRF for both synthetic graphs and the TUDataset graphs.
> >
> > **We observe that the run times for FoSR are almost an order of magnitude lower than that of SDRF.**
> >
> > We have also added new experiments in Appendix B.4 discussing the approximation error of FoSR (Figures 6 and 7).
> >
> > > Some naive baselines (i.e. fully connected graph at the last layer) and transformer-based methods are only mentioned but not included in the experiments.
> >
> > For our experiments we considered the current state-of-the-art methods SDRF and DIGL. The experiments we have provided clearly demonstrate the benefits of our strategy, which was the point of these experiments.
> >
> >
> > Also, as we mentioned in Section 5, our results are very competitive against results reported for other methods such as DiffWire (see Table 1 therein) that are using a different type of architecture that is not directly comparable to preprocessing methods like ours or the SDRF that sequentially adds edges. We report the exact numbers below.
> >
> > A. Arnaiz-Rodríguez, A. Begga, F. Escolano, and N. Oliver. DiffWire: Inductive graph rewiring via the Lovász bound. https://arxiv.org/pdf/2206.07369.pdf
> >
> > |    |   REDDIT-BINARY |   IMDB-BINARY |   COLLAB  |  MUTAG  |  PROTEINS |
> > |----------|:-------------:|:-------------:|:-------------:|:-------------:|:-------------:|
> > | FoSR     |  **$89.67 \pm 0.212$** | **$71.81 \pm 0.449$** | **$76.806 \pm 0.230$** | $86.150 \pm 0.761$ | $75.107 \pm 0.417$ |
> > | DiffWire |  $78.45 \pm 4.59$ | $69.93 \pm 3.32$ | $69.87 \pm 2.40$ | **$86.90 \pm 4.00$** | **$75.38 \pm 2.97$** |
> >
> >
> > We focus on the baselines that are most directly comparable to our method, but also agree that ultimately comparing with other methods and baselines is important. Hence, we have added rows to Table 1 containing results of making the last layer fully adjacent (+FA). Rewiring with FoSR still outperforms +FA in most cases when a relational architecture is used. We note that +FA is sometimes competitive, but this is highly dependent on the GNN architecture. In particular, it does not perform well for GCNs and R-GCNs. GNN architectures using transformers are also natural baselines and we will pursue these experiments when time permits.
> >
> > Thanks again for your valuable feedback. We hope our response and updated manuscript addresses your concerns and you consider raising your score.

---

> > > ### Comment · Reviewer_T9j4 · 2022-11-18
> > > **Re:**
> > >
> > > I thank the great effort of the authors for addressing my concerns. Most of them are well-addressed while a few of them remain (stated below). Nevertheless, I do appreciate the merits of the paper (mainly on the use of relational GNNs) and the improvement over rewiring-based methods. Hence I increase my rating to 6. Still, I hope the authors can seriously consider my concerns below in their future works.
> > >
> > > ### Are rewiring-based methods better than (graph) transformer-based methods or simple baselines?
> > >
> > > I notice that the authors add a fully connected graph (FA) as a baseline which is great. Please specify the FA details in the compared methods paragraph. I suppose it is only at the last layer? It is quite interesting to me that FA with R-GNNs can perform poorly in some cases (i.e., Reddit with R-GCN). I do not think the explanation provided by the authors (i.e., the two extreme cases) is convincing. In theory, shouldn't the model can learn the optimal weights between $\mathcal{E}_1$ and $\mathcal{E}_2$? Hence, merely considering the two extreme cases (using mostly only $\mathcal{E}_1$ or $\mathcal{E}_2$) is not convincing to me. Also, I do think these rewiring-based methods should compare to (graph) transformer-based methods, as they can resolve the same problem that we care about the most -- oversquashing. Nevertheless, I do believe that the FoSR method is the state-of-the-art rewiring-based method. I hope the authors can investigate in the future what is truly the best method for oversquashing, instead of considering only the rewiring-based methods.
> > >
> > > As a final remark, note that one benefit of FA is that the time complexity of applying it (i.e. construction) is negligible. Still, the authors already show that FoSR is indeed pretty efficient as well (i.e., Table 2). I also wonder what the performance of R-GNN with the choice of $\mathcal{E}_2 = \mathcal{E}/\mathcal{E}_1$ ***for all layers*** looks like. Actually, this is what I meant to be the simple baseline in the first place. I'm looking forward to checking it when the authors release their code.

---

> > > > ### Author Response · Authors · 2022-12-01
> > > > **Results for every layer FA**
> > > >
> > > > Thanks for your response and the clarification. We address some of your concerns about fully adjacent layers below.
> > > >
> > > > > I also wonder what the performance of R-GNN with the choice of $\mathcal{E}_2=\mathcal{E}/\mathcal{E}_1$ _**for all layers**_ looks like. Actually, this is what I meant to be the simple baseline in the first place.
> > > >
> > > > This is indeed a natural choice of rewiring. We have performed experiments for rewiring a graph making every layer fully adjacent. We report the results below.
> > > >
> > > > |       |                        |                    |                    |                    |                    |                    |
> > > > |-------|------------------------|--------------------|--------------------|--------------------|--------------------|--------------------|
> > > > |       |      REDDIT-BINARY     |     IMDB-BINARY    |        MUTAG       |       ENZYMES      |       COLLAB       |      PROTEINS      |
> > > > | GCN   | **$48.490 \pm 1.044$** | $48.170 \pm 0.801$ | $70.450 \pm 1.960$ | $18.333 \pm 1.038$ | $60.036 \pm 0.925$ | $51.798 \pm 0.419$ |
> > > > | R-GCN |   $49.950 \pm 0.593$   | $50.500 \pm 0.891$ | $70.500 \pm 1.836$ | $33.400 \pm 1.142$ | $71.670 \pm 0.882$ | $33.616 \pm 0.978$ |
> > > > | GIN   | $50.360 \pm 0.648$     | $49.160 \pm 0.870$ | $72.550 \pm 3.016$ | $28.383 \pm 1.052$ | $70.375 \pm 0.910$ | $32.894 \pm 0.390$ |
> > > > | R-GIN   | $56.855 \pm 0.943$ | $71.480 \pm 0.876$ | $83.050 \pm 1.518$ | $54.950 \pm 1.331$ | $71.045 \pm 0.909$ | $75.432 \pm 0.475$ |
> > > >
> > > > Comparing these numbers to Table 1, we see that making every layer fully adjacent does not improve performance on average compared to making only the last layer fully adjacent, and often hurts performance substantially. This is in line with the observations by Alon and Yahav [1] that making every layer fully adjacent harms GNN performance.
> > > >
> > > > > It is quite interesting to me that FA with R-GNNs can perform poorly in some cases (i.e., Reddit with R-GCN). I do not think the explanation provided by the authors (i.e., the two extreme cases) is convincing. In theory, shouldn't the model can learn the optimal weights between $\mathcal{E}_1$ and $\mathcal{E}_2$? Hence, merely considering the two extreme cases (using mostly only $\mathcal{E}_1$ or $\mathcal{E}_2$) is not convincing to me.
> > > >
> > > > As you rightfully point out, the GNN is capable of learning the optimal weights between the original edges and rewired edges. So in principle, any rewired GNN should be able to outperform an equivalent GNN without rewiring. However, this optimal combination varies depending on our choice of $\mathcal{E}_2$. If we use an extreme choice of $\mathcal{E}_2$ such as including no edges or including all edges, it becomes less helpful as a feature regardless of how much we weight it.
> > > >
> > > > > Please specify the FA details in the compared methods paragraph.
> > > >
> > > > Thanks for catching this. We will add the following brief description of FA layers to the final version.
> > > >
> > > > Fully adjacent layers [1] rewire the graph by adding all possible edges, setting $\mathcal{E}_2 = (\mathcal{V} \times \mathcal{V}) \setminus \mathcal{E}_1$. We include results for rewiring only the last layer (last layer FA) and rewiring every layer (every layer FA).
> > > >
> > > > [1] U. Alon and E. Yahav. On the bottleneck of graph neural networks and its practical implications. In
> > > > International Conference on Learning Representations, 2021

---

### Official Review · Reviewer_2pg5 · 2022-10-22

**Confidence:** 3
**Correctness:** 4
**Technical Novelty And Significance:** 3
**Empirical Novelty And Significance:** 3
**Recommendation:** 8

**Clarity, Quality, Novelty And Reproducibility:**

Clarity:

The paper is overall rather clear, apart from a few mathematical points that are a bit heavy for someone unfamiliar with the Graph Laplacian and the notion of spectral gap.

I found section 4 easier to read and understand (in the details) than section 3.

I think the paper would benefit a lot from a few more "intuitive" explanations. For instance, I feel like the spectral gap, i.e. the value of the second eigenvalue of the normalized Graph Laplacian, should somehow relate with the second eigenvalue of (normalized) A: I know that the first mode of A corresponds to the stationary distribution of the Markov Chain associated to A. The second mode (eigenvector) of A corresponds to the first excitation of this stationary probability distribution, and the eigenvalue relates to the associated timescale (with a relation like tau = - log(lambda_2). Look at "Perron Cluster Cluster Analysis" for details on these ideas (not mine !)

Relating these ideas with the spectral approach could increase the readership a lot (or increase the fraction of happy readers).

Other point that would need more intuitive explanations, for me at least: the definition 2: why is there a sup ?

Also the end of section 4 (last paragraph, bottom page 7) is mysterious and would need more explanations (it is said that one can "often" compute the minimal value... -> what if often ? What are the conditions ? (and why can we replace a full search over pairs by single index-searches ? -- I didn't get it).

Top of page 8: why can we relax some conditions ? This paragraph was really not obvious to me. Probably just a bit more discussion would enlighten me.


Quality:

- the paper seems sound and correct. I did not re-run experiments. I checked the maths only to some extent. I found a few typos, by the way (see lower).

- The paper is well introduced and situated within literature: I did not know of oversquashing, and now I (think I) know about it.

- The background on spectral graph theory is ok, could provide a few more intuitive views, e.g. in terms of Markov chains (random walk on the graph).


In terms of novelty.

- Disclaimer: I am not aware of the rest of the literature on oversquashing (neither oversmoothing, but it's less relevant). I have to believe that the experiments, esp. the comparison with other SOTA methods, were done in good faith.

- After a very quick literature search, however, it seems the paper cites the recent relevant works.

- I did not find other recent rewiring methods than the 2 this papers compare to. I could have missed one.


Reproducibility:
- the code is available, I haven't run it but is seems well organized.
- Here I have a question:
    I would like to know how the iteration counts presented in table 3, used for experiments, where chosen.
     Was it by checking out figure 1-style results, i.e. a form of saturation of the spectral gap with added edges, or was it optimized for performance ?

----------
Clarity: improvements suggestions in detail

Although GNN models are presented in sec. 2.2, it is unclear to me what are the models presented in experiments.
The GIN acronym is never defined.
I think authors should make this much clearer, as it's a key point, and for now it's quite confusing.
In the same line of comments, I think they should stress out where the different nature of rewired (Added) edges appears in the GNN architecture, how it is dealt with. So far, I understand it just comes as a separate channel, but is mixed (added with learned weights) with the regular edges). It seems almost too simple !



------------
typos:
in proof of theorem 3, 3rd equation:
- an X becomes (1-alpha)X ? One could just choose Theta=(1-alpha).I and get this (or I missed something).
- last =, = (I - alpha L), an X at the right is missing.
Also, I guess it's obvious that A=A1+A2, but maybe you could mention it ?

Before proposition 5: you assume symmetric matrices. So you deal only with undirected graphs, correct ? This should be mentioned/discussed.





**Strength And Weaknesses:**

Weaknesses:

- Section 3 is hard to read, and contains some general theorems in which the proof is in a sense "disappointing" (theorem  3): the weight matrices \Theta used in the proof are trivial (proportional to identity)
- The experiments are described a bit too succinctly, making it hard to infer exactly how experiments were performed (more details later in the review).

Strengths:
- the two contributions seem to be genuinely good ideas, with practical added value: the special label for added edges seems to indeed improve accuracy.
- The rewiring algorithm is not too expensive (actually seems pretty cheap, under some conditions), and also clearly works well (better than the two previous method it's compared to).

**Summary Of The Paper:**

This paper deals with oversquashing in GNNs, that is, when information flows between nodes (receptive field large enough), but is too compressed (squashed) into the (finite) features to be correctly exploited for prediction (because number of neighbors grows exponentially with radius of receptive field).

The paper has 2 contributions:

- First a key idea, which is that in a rewiring procedure, one can label the added edges differently from the original ones, so that the GNN knows they are "fake edges", only here to help propagate information.

- Second, based on spectral gap/graph Laplacian mathematical arguments, they propose a new rewiring algorithm. The rewiring algorithm is inspired from the arguments, but does not guarantee the best rewiring (largest increase in spectral gap) at each step: instead it is approximate, so as to be reasonably cheap (computationally).

Both these contributions are backed by experiments, comparing with 2 other rewiring methods.

**Summary Of The Review:**

The paper is overall good, and if the ideas presented are indeed new, should be published. The 2 contributions seem to be directly applicable, and yet are avenues for future work (improved architectures that well exploit the added edges with new label, and other approximations or schemes for adding edges, in the spirit of increasing the spectral gap).

Technically and empirically, contributions are somewhat new, in that the rewiring and the way to process are indeed new, but the very idea of rewiring, and the very idea of fighting oversquashing, are not themselves new.

---

> ### Author Response · Authors · 2022-11-17
> **Initial response to Reviewer 2pg5**
>
> Thanks for your valuable feedback and suggestions. We have incorporated all your suggestions as detailed below.
>
> > I found section 4 easier to read and understand (in the details) than section 3.
>
>
> We have now updated Sections 3.1 and 3.2 with more explanations as suggested. Please see the text highlighted in blue therein.
>
>
> > I think the paper would benefit a lot from a few more intuitive explanations ... The background on spectral graph theory is ok, could provide a few more intuitive views, e.g. in terms of Markov chains (random walk on the graph) ... Relating these ideas with the spectral approach could increase the readership a lot (or increase the fraction of happy readers).
>
>
> Thanks for the suggestion! For the intuition behind our idea of spectral gap optimization, we have now added more details to the theoretical discussion in Appendix A, where we explain why the spectral gap is a well known and useful quantity to measure structural bottlenecks in an input graph and hence is a good candidate for measuring oversquashing. As per your suggestion, we also briefly comment on how having a large spectral gap implies that a simple random walk on graph is rapidly mixing (see text highlighted in blue in Appendix A.1).
>
>
> > Other point that would need more intuitive explanations, for me at least: the definition 2: why is there a sup?
>
>
> Thanks for the comment. We have now added an explanation of this in the main text following Definition 2. We take a supremum in the numerator to measure the largest possible relative change in the Dirichlet energy, and a supremum in the denominator to measure the largest possible rate at which $\varphi$ can scale up the norm of $X$. Intuitively, if the supremum in the numerator is small, the Dirichlet energy will decay substantially regardless of the value of $X$. By examining these extreme cases, we are able to get a theoretical handle on the smoothing properties of $\varphi$.
>
> > Also the end of section 4 (last paragraph, bottom page 7) is mysterious and would need more explanations (it is said that one can "often" compute the minimal value... $\to$ what if often ? What are the conditions ? (and why can we replace a full search over pairs by single index-searches ? -- I didn't get it). Top of page 8: why can we relax some conditions ? This paragraph was really not obvious to me. Probably just a bit more discussion would enlighten me.
>
>
> We have added a brief explanation to the paragraph for clarity. The argmin and argmax check is a simple heuristic that we use to speed up the search for a minimal value of $y_iy_j$. The idea is that if we have a set of positive and negative values $y_1 \leq \cdots \leq  y_n$, the smallest possible product we can form will be $y_1y_n$. This heuristic is not a crucial part of our algorithm, but it is likely to work when the graph is sparse, since the proposed edge $(i, j)$ is less likely to already be in the graph. We have clarified this in the writing of the paragraph "Computational complexity" in P.8 (please see the highlighted text in blue).
>
>
>
> > The GIN acronym is never defined. I think authors should make this much clearer, as it's a key point, and for now it's quite confusing.
>
> Thanks for pointing this out. We have now defined GINs in Section 2.2.
>
>
> > I think they should stress out where the different nature of rewired (Added) edges appears in the GNN architecture, how it is dealt with. So far, I understand it just comes as a separate channel, but is mixed (added with learned weights) with the regular edges). It seems almost too simple!
>
>
> Thanks for this comment. We have now added a more explicit description in Section 3.1 clarifying the point (please see the highlighted text in blue therein).

---

> > ### Comment · Reviewer_2pg5 · 2022-11-28
> > **Acknowledgement of answers**
> >
> > I thank the authors for addressing my questions, and thus clarifying some points.
> > I see a good deal of improvement has been done in the paper, and in expanding the appendix.
> > Reading other referee's reports (and author's answers), I've also learned some things.
> >
> > My score was already high so I do not change it, but I think this discussion was fruitful.
> >
> > Given the importance of Fig. 5 , it's a bit of a pity that :
> > - it has so few points. Btw markers should be added to lines, as here only few actual measurement points are available (I guess from the line's change in slope).
> > - it's a pity that it is not directly in the main text ! (but maybe the space constraint does not allow it)
> > I see B.3 is referred to in the discussion, at least there's that.
> >
> > Sincerely,

---

### Official Review · Reviewer_j5dr · 2022-10-24

**Confidence:** 3
**Correctness:** 3
**Technical Novelty And Significance:** 3
**Empirical Novelty And Significance:** 2
**Recommendation:** 8

**Clarity, Quality, Novelty And Reproducibility:**

The article is well written and clear. The code is available and the artile appears to be reproducible.

The novelty is fair, yet incremental in some aspects. Maybe the novelty and improvement in performance (Empirical Novelty And Significance) would be rated stronger if the weaknesses 2 and 3 + the minor questions are addressed satisfactorily.

**Strength And Weaknesses:**

Strength:

1- The idea and derivation of the first order spectral rewiring is very well explained in Section 4 (even if it relies on known tools or algorithms, the idea is, for me, new in this context).

2- The resulting method shows improvements in effeiciency as compared to contenders, and the numerical experiments are extensice enough to assess that.

3- Globally the article is clear and it's a nice read, without any superfluous elements.

Weaknesses

1- The issue addressed, navigating the trade-off between oversquashing and oversmoothing, exists yet I am not certain it is a major one and several works have already discussed and addressed the issue. The contribution is, on this side, incremental.

2- The algorithm relies on approximate (perturbation-based + power-method eigenvectors computation + approximation of argmin (y_i y_j) which appear to work yet are not really controled or checked. What would happen with difference choices of k, r and of the argmin method ? Are the results robust ? What would be the choice of r if k has to be increase ?

3- there don't appear to gave results comparing the computation time to methods from contenders. Also, parameters (as in D.1) could be varied a little bit, at least to check robustness against ther variation.

 Minor questions :

- results on ENZYMES appear to be low as compared to what I have seen in other works; Why that ?

-   the number of digits in the numerical results (5 digits on percentages) appear to be high. Is it really trustable ? Even with 100 runs, I wold expect less precision of the accuracy results.

- the datasets are rather small size (in average number of nodes and/or number of graphs). Given the  dicussion about computational complexity and the approximations leveraged in the method, I would have expected results on larger datasets as well.

- I don't think that Appendix A is useful (I think that expected readers are familiar with that).

- Figure 1 should be larger so as to be more readable.

**Summary Of The Paper:**

The work addresses the issue of oversquashing in GNN in proposing a new approach of rewiring graphs so as to improve the spectral gap of the graph (as it has been studied that spectral gap is a key feature in preventing oversquashing). The proposed rewiring algorithm comes from a study of perturbation of the Fiedler vector  and spectral gaps when adding one edge. The methods is conceptually simple and relevant and some numerical results show its performance.

**Summary Of The Review:**

My recommandations is for marginal accept, because it is a sound work on an existing issue, even if the work is in part incremental.


Update after revision: I increase the score to 8: Accept.

---

> ### Author Response · Authors · 2022-11-17
> **Initial response to Reviewer j5dr (part 1)**
>
> Thanks for your valuable feedback and suggestions. We have incorporated all your suggestions as detailed below.
>
>
> > The issue addressed, navigating the tradeoff between oversquashing and oversmoothing, exists yet I am not certain it is a major one and several works have already discussed and addressed the issue. The contribution is, on this side, incremental.
>
> There is indeed a prior literature on oversquashing and oversmoothing as we also point out in the introduction. However, we are not aware of **any** work talking about the "tradeoff" between oversquashing and oversmoothing. While the two issues have been recognized, the identification of approaches to effectively solve them is still an important ongoing effort in the community. If the reviewer knows of other approaches addressing the tradeoff between oversquashing and oversmoothing, we kindly request that they point at concrete references. To better highlight this tradeoff that motivates our proposed methods, we significantly expanded the discussion in Appendix B.3. Concretely:
>
> - Input graphs with small spectral gaps are prone to oversquashing, whereas input graphs with large spectral gaps are prone to oversmoothing. We use R-GNNs to alleviate oversmoothing, and Table 3 shows that **relational architecture indeed helps to reduce oversmoothing**, since the relational rewiring leads to higher Dirichlet energies.
>
> - While relational GNNs can help with reducing oversmoothing, there is an optimal number of edges to add, beyond which performance declines. Figure 5 shows that the performance for R-GCNs peaks at around 25 iterations, which is reflected in both the test accuracy and Dirichlet energy of the final layer.
>
> > The algorithm relies on approximate (perturbation-based + power-method eigenvectors computation + approximation of $\textrm{argmin} (y_i y_j)$ which appear to work yet are not really controlled or checked. What would happen with difference choices of $k, r$ and of the argmin method? Are the results robust? What would be the choice of $r$ if $k$ has to be increase?
>
>
> For the **choice of $k$**, the number of rewiring iterations $k$ must be chosen to balance the tradeoff between oversquashing and oversmoothing. If $k$ is too high, we will experience oversmoothing, and if $k$ is too low, we will experience oversquashing. We empirically verify that this tradeoff exists in Figure 5 (in Appendix B.3), which shows that the performance for R-GCNs peaks at around 25 iterations, which is reflected in both the test accuracy and Dirichlet energy of the final layer.
>
>
> For the **choice of $r$**, we note that $r$ determines the initial number of power iterations, and is not related to $k$. We only use it to provide an initial estimate of the spectral gap and associated eigenvector. We used $r=5$ for our experiments on downstream GNN tasks. In principle, if the Laplacian of a graph is ill-conditioned one could choose a higher value of $r$.
>
>
> The **argmin and argmax check** is a simple heuristic that we use to speed up the search for a minimal value of $y_iy_j$. It is not a crucial part of our algorithm, but it is likely to work when the graph is sparse, since the proposed edge $(i, j)$ is less likely to already be in the graph. We have clarified this in the writing of the paragraph "Computational complexity" in P.8.
>
>
>
> > There don't appear to gave results comparing the computation time to methods from contenders. Also, parameters (as in D.1) could be varied a little bit, at least to check robustness against ther variation.
>
> - **Computation time**: We now include a new Appendix B.2 comparing the computation time of FoSR with the state-of-the-art method SDRF. Table 2 compares the computation time of FoSR and SDRF on the TUDataset. Figure 4 compares their run time on Erdos-Renyi graphs of various sizes. For both, we observe that the run times for FoSR are almost an order of magnitude lower (i.e., faster) than that of SDRF.
>
> - **Variation of parameters**: This is a valid point, and indeed, the values reported in Table 1 are only for the optimal rewiring iteration counts. We have also now included Figure 5 in Appendix B.3 showing explicitly the tradeoff between oversquashing and oversmoothing with respect to the rewiring iteration. We have also added new experiments in Appendix B.4 discussing the approximation error of FoSR (Figures 6 and 7).

---

> > ### Author Response · Authors · 2022-11-17
> > **Initial response to Reviewer j5dr (part 2)**
> >
> > > results on ENZYMES appear to be low as compared to what I have seen in other works; Why that?
> >
> > One possible reason for this might be that we use same hyperparameters (aside from rewiring iterations, such as hidden dimension, learning rate, and dropout) across the board for all the datasets in Table 1. We did not tune these hyperparameters since our goal was to compare FoSR to existing rewiring methods for a fixed choice of GNN architecture. We will seek to pursue hyperparameter tuning for individual tasks if time permits.
> > >  Figure 1 should be larger so as to be more readable.
> >
> > We have increased the font size to improve the readability of Figure 1.
> >
> > > the number of digits in the numerical results (5 digits on percentages) appear to be high. Is it really trustable ? Even with 100 runs, I wold expect less precision of the accuracy results.
> >
> > Indeed, the reported values are for 100 runs. Please note that we are not making a statement on the significance of the last digits of our accuracies; we are simply reporting our sample means and standard errors.
> >
> > > the datasets are rather small size (in average number of nodes and/or number of graphs). Given the dicussion about computational complexity and the approximations leveraged in the method, I would have expected results on larger datasets as well.
> >
> > Despite the datasets being small, we compare our FoSR with existing methods like SDRF that take a significant amount of computational resources; see Table 2 and Figure 4 in Appendix B.2. The extremely high compute/memory resource requirements of SDRF is also reported in Section 5.1 in Bober et al. (2022).
> >
> >    J. Bober, A. Monod, E. Saucan, and K. N. Webster. Rewiring networks for graph neural network training using discrete geometry, 2022 (https://arxiv.org/abs/2207.08026)
> >
> > > I don't think that Appendix A is useful (I think that expected readers are familiar with that)
> >
> > We thank you for this comment. However, we felt that all the other reviewers had in fact suggestions to include more intuition and motivation for our spectral gap optimization approach. In light of this, we have now modified Appendix A to highlight why the spectral gap is an important quantity for quantifying structural "bottlenecked-ness" of graphs and addressing oversquashing in GNNs.
> >
> >
> > We thank you for your highly valuable feedback and hope that our response and updated manuscript addresses all of your concerns and you consider raising your score.

---

> > > ### Comment · Reviewer_j5dr · 2022-11-18
> > > **Acknowledging revision**
> > >
> > > Dear authors,
> > >
> > > I have read your answers and the changes made. The proposed article is indeed improved that way (and I acknowledge also the interests of the improved insights about your choices in Appendix A). I will increase my score based on this revisions.
> > >
> > > Sincerely.

---

### Official Review · Reviewer_42Gr · 2022-11-02

**Confidence:** 4
**Correctness:** 3
**Technical Novelty And Significance:** 3
**Empirical Novelty And Significance:** 3
**Recommendation:** 8

**Clarity, Quality, Novelty And Reproducibility:**

- Clarity: The writing of the paper is high quality. It reads well and it does a great job at conveying the key ideas. Some additional discussions would help round out the motivation and make the paper more self-contained, but overall it is well-written.

- Novelty: The main contribution of the paper is novel and original. It has multiple elements: a new graph rewiring approach called FoSR, an associated relational GNN formalism and corresponding theoretical results.

- Reproducibility: The paper includes a reproducibility statement, and code is provided to aid in it. I have not run the experiments by myself.

**Strength And Weaknesses:**

### The paper has several strengths:
- It is remarkably well-written and the exposition is very clear.
- The proposed techniques are elegant and innovative, but mostly nontrivial.
- Theoretical results hint at the benefits of the approach.
- Detailed experiments demonstrate the benefit of both techniques, individually and also when combined.

### Notable weaknesses:
Overall, there are some missing points in the discussion which make me a bit unsure about this paper on a second read. Depending on how these points are addressed, I can be incentivized to modify my rating in either direction.

- Missing discussion about spectral gap maximization.

The paper does a good job at conveying the core ideas from related work without overloading the reader. One thing I was missing is an overview of previous results for why graph rewiring via spectral gap maximization is a worthwhile approach for alleviating oversquashing. Since this seems to be the foundation on which the corresponding FoSR algorithm is built, I think it would go far if there was an intuitive overview of the core idea.

- Theorem 3 does not tell the whole story.

Theorem 3 seems to be a useful result regarding the flexibility of R-GNN. However, I have some doubts whether it tells the full story. Specifically, when the weight matrices $\Theta_1$ and $\Theta_2$ are reduced towards zero, the rate of smoothing reduces as the contribution from the direct and indirect neighbours decrease. In the corner case of $\Theta_1 = \Theta_2 = 0$, only the self-loops remain with the R-GNN becoming a node-wise linear transformation, and this setting also clearly achieves minimal oversmoothing. However, the structural information is lost in this case as well. In this way, Theorem 3 does not tell the whole story because somehow we are trying to maintain propagation along the important edges while maintaining a low rate of smoothing. It seems there is a missing component here to really guarantee the strength of these R-GNNs in practice. It would be interesting to know more about how this interaction happens with the graph rewiring approach utilized.

- Experiments do not report investigation regarding the realized spectral gap and associated trade-offs.

The experiments demonstrate improved downstream performance after applying the proposed rewiring approach. On the other hand, no investigation of robustness is carried out in terms of the realized spectral gap itself. I would be curious about the following comparison for the compared rewiring methods: number of rewiring iterations - realized spectral gap - downstream performance. In this case, other model hyperparameters could be treated as fixed. This also connects to the previous point, as it looks like simply maximizing the spectral gap is not sufficient (otherwise we could just choose a larger number of rewirings). Although the R-GNN can learn to reduce oversmoothing, it seems like it's main method for doing so is to effectively eliminate the newly introduced connections, hence making the R-GNN more similar to classic GNN. It seems like we are still somehow trying to balance both phenomenons (oversmoothing and oversquashing) with some added flexibility. For example, how this trade-off happens with respect to the rewiring steps could be investigated and explained in more detail empirically.

- Robustness of FoSR approximation is unexplored.

During the derivation of FoSR, several involved calculations are carried out where the spectral gap is approximated (first the order-1 Taylor expansion, and then an asymptotic argument to discard some of the terms from the maximand). Consequently, as with most approximations, there is some error incurred. At the moment, there is no intuition about when (just from the point of view of spectral gap maximization) this rewiring approach is expected to work well, or if it could fail in certain cases. Some discussion would be interesting here.

- Motivation of tools utilized for oversmoothing analysis.

Section 3 first sets the foundation for the theoretical analysis by introducing a carefully chosen set of tools. I found the associated references to be somewhat scarce. The Dirichlet energy associated with the graph is referenced, but I did not see references for the associated rate of smoothing "metric". Is this something that the authors propose themselves, or was this used in previous works? In the former case, I think it would be worthwhile to provide a more detailed justification about why this is a good quantity to measure oversmoothing.


POST-REBUTTAL: Rating increased to 8 in view of significant improvements.

**Summary Of The Paper:**

The paper investigates the oversquashing phenomenon in Graph Neural Networks from the perspective of graph rewiring methods. The purpose of graph rewiring is to alleviate structural bottlenecks and facilitate the flow of information between more distant nodes. The contribution of the paper is twofold: 1) for any graph rewiring approach, a relational GNN implementation is proposed, which is able to make use of the augmented edge set without forsaking the information contained in the graph topology; 2) First-Order Spectral Rewiring is proposed as an efficient rewiring mechanism, which provides a principled way for selecting edges by maximizing the change (up to first-order approximation) in the spectral gap. In support of 1), a theoretical result is provided, which seems to hint that the proposed R-GNN formalism might be able to make use of the augmented index set without exacerbating a related phenomenon called oversmoothing. Experimental results demonstrate the benefit of both proposed techniques compared to non-relational GNNs and alternative rewiring methods.

**Summary Of The Review:**

The paper investigates oversquashing and oversmoothing. It communicates the ideas particularly well. The proposed techniques (R-GNN and FoSR) are novel and original. Some theory is provided to motivate the R-GNN with some open questions at the moment. Experiments demonstrate the benefit of the both techniques with interesting results compared to alternative rewiring approaches and non-relational GNN. Discussion of trade-offs and some theoretical questions would help strengthen the message. As mentioned above, I am looking forward to the authors' rebuttal regarding my questions, which may affect the final rating (in both ways).

---

> ### Author Response · Authors · 2022-11-17
> **Initial response to Reviewer 42Gr (part 1)**
>
> Thanks for your valuable feedback and suggestions. We have incorporated all your suggestions as detailed below.
>
> > Missing discussion about spectral gap maximization ... I think it would go far if there was an intuitive overview of the core idea.
>
> Thanks for your valuable suggestion. For the intuition, we have now added details to the discussion in Appendix A, where we explain why the spectral gap, which is a well known quantity in the literature in relation to random walks on graphs, is a useful quantity to measure structural bottlenecks in an input graph and hence is a good candidate for measuring oversquashing.
>
>
> The intuitive explanation of the core idea is the following:
> When the spectral gap is high, then the graph has no structural bottlenecks in the sense that every part of the graph is connected to the rest of it by a large fraction of its edges. This implies better information propagation and in turn less oversquashing.
>
>
> We have also added in Appendix B several experiments addressing the tradeoff between overquashing and oversmoothing, the computation time, and the rate of spectral expansion. Please see our "Common response: Summary of main updates to the manuscript".
>
>
>
> > Theorem 3 does not tell the whole story.
>
>
> As you rightly pointed out, if $\Theta_1 = \Theta_2 = 0$, all node connectivity is forgotten. Indeed, this is an extreme case where the R-GNN does not smooth out the node features at all, which is not desirable. In addressing oversquashing and oversmoothing, we wish for R-GNN layers to achieve a rate of smoothing which is neither too high nor too low. However, as a theoretical tool, it is useful to show that the R-GNN can smooth with maximal and minimal rates. Then, we interpolate between these rates, showing that the R-GNN can achieve any rate of smoothing in between. We do not intend to suggest that GNNs use weights of 0 for $\Theta_1$ and $\Theta_2$ in practice. When $\Theta_1 = \Theta_2 = 0$, we are in a degenerate case where the Dirichlet energy is 0s. The other extreme case is where we use no self loops ($\Theta = 0$), and in that case the graph is maximally smoothed. We can hit any level of smoothing in between, which is what we want the GNN to achieve in practice.
>
>
> In our proof, we have a parameter $\alpha \in [0,1]$, which can certainly take non-degenerate values. This corresponds to scaling up and down the relative importance of the adjacency matrix versus self-loops in feature updates. It is true that we only require diagonal matrices to show that these rates of smoothing are possible. This shows that the ability of an R-GNN to control the relative sizes of weights between classes is useful to control oversmoothing. This fact is not necessarily specific to diagonal matrices, but it is easiest to establish without more assumptions in this case.
>
> > Experiments do not report investigation regarding the realized spectral gap and associated tradeoffs ... It seems like we are still somehow trying to balance both phenomenons (oversmoothing and oversquashing) with some added flexibility. For example, how this tradeoff happens with respect to the rewiring steps could be investigated and explained in more detail empirically.
>
>
> Indeed, it is a natural question to ask how downstream performance is affected by changes in the spectral gap, which in turn is determined by the number of rewiring iterations. We have now included a **new appendix (Appendix B.3) expanding on the discussion on the tradeoff between oversquashing and oversmoothing**.
>
> - Input graphs with small spectral gaps are prone to oversquashing, whereas input graphs with large spectral gaps are prone to oversmoothing. We use R-GNNs to alleviate oversmoothing, and Table 3 shows that **relational architecture indeed helps to reduce oversmoothing**, since the relational rewiring leads to higher Dirichlet energies.
>
> - While relational GNNs can help with reducing oversmoothing, there is  an optimal number of edges to add, beyond which performance declines. Figure 5 shows that the performance for R-GCNs peaks at around 25 iterations, which is reflected in both the test accuracy and Dirichlet energy of the final layer.

---

> > ### Author Response · Authors · 2022-11-17
> > **Initial response to Reviewer 42Gr (part 2)**
> >
> > > Robustness of FoSR approximation is unexplored ... Consequently, as with most approximations, there is some error incurred. At the moment, there is no intuition about when ... this rewiring approach is expected to work well, or if it could fail in certain cases. Some discussion would be interesting here.
> >
> >
> > This is a valid point and we have added **new experiments in Appendix B.4** discussing the approximation error of FoSR (Figures 6 and 7).
> >
> >
> > When rewiring a graph using FoSR, the objective at each iteration is to add the edge which maximally increases the spectral gap. Since FoSR makes this calculation using a first-order approximation of the spectral gap, there is some error incurred between the estimated spectral gap and the actual spectral gap. This error is caused by two factors. The first is the error from using the first-order change of the spectral gap instead of the actual change. The second error accrues from using the dominant term of the first-order change instead of the first-order change. We record each of these three quantities (actual change, first-order change, FoSR approximated change) for graphs in the ENZYMES dataset in Figure 6. We see that the actual change is well-correlated with the approximations, and that the approximations are well-correlated with each other.
> >
> >
> > While FoSR accumulates some error from the actual change in the spectral gap, we are primarily interested in how this affects its ability to choose good edges for rewiring. Hence, it is natural to ask how FoSR compares to a greedy algorithm which computes the spectral gap exactly. We test the performance of FoSR on a synthetic dumbbell graph, consisting of two cliques of size 50, connected via a path of length 3 (see Figure 7). We see that the spectral evolution of FoSR is indistinguishable from that of the greedy method, indicating that the approximation is strong and that the two methods choose similar edges.
> >
> >
> > Optimizing only the first-order change gives our method a significant computational advantage over competing methods such as the SDRF. In Appendix B.2, we show that FoSR has significantly smaller computation overhead compared to SDRF.
> >
> > > Motivation of tools utilized for oversmoothing analysis ... I did not see references for the associated rate of smoothing "metric" ... provide a more detailed justification about why this is a good quantity to measure oversmoothing.
> >
> >
> > Yes, we propose this rate of smoothing, and are not aware of other works which use the same metric. We have now added a brief explanation of our motivation behind this definition in the main text (see page 5 in Section 3.2). The intuition behind our definition is most transparent when the mapping in consideration is linear. The Dirichlet energy is a quadratic form that captures variation in node features, so it is natural to consider the operator norm squared of a matrix with respect to this quadratic form. Intuitively, if the supremum in the numerator is small, the Dirichlet energy will decay substantially regardless of the value of $X$. We also want our notion of rate of smoothing to be scale-invariant: Multiplying all entries by a scalar should not change the rate of smoothing of a mapping. To impose this invariance, we divide by the operator norm squared of the matrix with respect to the usual $2$-norm. We use this metric to quantify the flexibility of R-GCNs. This would be trivial in the absence of scale-invariance, since we could just scale up the weights of the matrices, but this would not address the oversmoothing issue. Therefore, our normalization factor in the denominator of the rate of smoothing is essential. Hence, **we believe that our definition of the rate of smoothing fundamentally captures the idea of oversmoothing**.
> >
> >
> > We have made significant efforts to address your concerns. Based on your suggestions, we have now included several new experiments in our revision, which in our view substantially strengthened the paper. We hope you will agree with this and consider raising your score.

---

> > > ### Comment · Reviewer_42Gr · 2022-11-21
> > > **Acknowledgement of rebuttal**
> > >
> > > Thanks to the authors for the interesting discussions. My concerns were reasonably addressed. I find the additional experiments in Appendix B very intriguing. The added description regarding the relevance of spectral gap and the rate of smoothing seems clear. The paper is state-of-the-art within its own niche, that is, addressing oversquashing via rewiring techniques in the context of (semi-)isotropic GNNs. Indeed, it is of crucial importance to understand this case before moving onto anisotropic (attentional) GNN models. Overall, the rebuttal has significantly improved my view of the paper, which I believe is interesting in its own right to a wide range of the community, so I've raised my score.
> > >
> > > Finally, the proposed relational mechanism also seems to be potentially useful in anisotropic models, such as GAT and graph transformers, and I suggest to the authors to also explore this direction in future work.

---

### Author Response · Authors · 2022-11-17
**Common response: Summary of main updates to the manuscript**

We thank all the reviewers for their careful consideration of our manuscript and thoughtful suggestions. We have made significant improvements to our initial results in response to their comments. We report several additional experiments in Appendix B. We summarize the main updates below:

* For the **motivation and intuition behind our idea of spectral gap optimization**, we have now added more details and intuition in Appendix A, where we provide empirical intuitions and explain why the spectral gap is a well known and useful quantity to measure structural bottlenecks in an input graph and hence is a good candidate for measuring oversquashing, and in Appendix B, where we provide empirical intuitions.


* We have added **new experiments in Appendix B.1** for TUDataset (Figure 3) showing that FoSR has a **faster rate of spectral expansion** compared with existing methods.


* We have added **new experiments in Appendix B.2** showing that optimizing only the first-order change in the spectral gap gives FoSR a **significant computational advantage** over SDRF for the TUDataset graphs (Table 2) as well on a synthetic dataset consisting of Erd\"os-R\'enyi graphs (Figure 4).


* We have added **new experiments in Appendix B.3** showing that the **relational architecture indeed helps to reduce oversmoothing**, since the
relational rewiring leads to higher Dirichlet energies (Table 3). While relational GNNs can help with reducing oversmoothing, there is a tradeoff between oversquashing and oversmoothing as a function of the number of rewiring iterations. Figure 5 shows this tradeoff by way of the optimal number of edges to add, beyond which performance declines.


* We have added **new experiments in Appendix B.4** discussing the approximation error of FoSR, showing that it is well-behaved (Figures 6 and 7).


All modifications/updates in the manuscript taking care of the reviewer suggestions are highlighted in blue.


Kindly let us know if you have any further questions. We remain attentive to your comments and feedback!

---

### Decision · Program_Chairs · 2023-01-20

**Decision:**

Accept: poster

**Justification For Why Not Higher Score:**

The method is essentially a heuristic, but that seems to work.

**Justification For Why Not Lower Score:**

The reviewers seem to deem the evaluation sufficient and the method novel enough to warrant acceptance.

**Metareview: Summary, Strengths And Weaknesses:**

__Summary (for meta review)__ This paper addresses the problem of oversquashing. At a high level, the core of the paper is the proposal of a computationally efficient algorithm that prevents oversquashing by adding edges to the graph. The criterion for adding edges is based on a spectral expansion that aims to minimize the second order eigenvalue of the matrix --- thereby increasing the spectral gap. Their approach gives special labels to added edges so that the new edges are treated differently from the original ones: the authors propose using a relational GNN on the induced “multigraph”. This allows to preserve the input graph topology while using the new edges to improve its connectivity. More precisely, the authors build upon the concept of Dirichlet energy to quantify the amount of smoothing (the higher, the more smoothing is induced by the GNN). The authors perform a linear (Taylor) expansion of the effect of an edge addition on the second eigenvalue of the matrix . They find that this linear expansion is a sum of three terms. For computational reasons, they suggest using the first of these terms (which dominates if the degrees are comparable in size) to maximize this term.

__Summary of the reviews__ The reviewers were quite enthusiastic about the submission. After initial reserves about the justification of the proposed solution (increasing the spectral gap), and the trade-off between oversquashing and oversmoothing, the reviewers seemed universally satisfied with the additional explanations provided by the authors and included in the appendix. The method seems to work fairly well. Concerning the weaknesses of the approach: the theorem provided by the authors to explain how their method allows to achieve lower Dirichlet energy than GNNs seems a little off. The authors also do not discuss the limitation nor try to explore the importance of the approximation they are making to in their approximation of the perturbation of the second eigenvalue.  Consequently, the proposed method was deemed novel enough to deserve being accepted at ICLR, but not as a spotlight.

**Note From Pc:**

if the above contains the word "oral" or "spotlight" please see: "oral" presentation means -> notable-top-5% and "spotlight" means -> notable-top-25%. As stated in our emails, we are disassociating presentation type from AC recommendations

**Summary Of Ac-Reviewer Meeting:**

N/A